# Hypothalamic corticotropin-releasing hormone neurons modulate sevoflurane anesthesia and the post-anesthesia stress responses

**Shan Jiang[†], Lu Chen[†], Wei-Min Qu, Zhi-Li Huang\*, Chang-Rui Chen\***

Department of Pharmacology, School of Basic Medical Sciences; State Key Laboratory of Medical Neurobiology and MOE Frontiers Center for Brain Science, Institutes of Brain Science, Fudan University, Shanghai, China

**\*For correspondence:**
huangzl@fudan.edu.cn (Z-LH);
changruichen@fudan.edu.cn
(C-RC)

[†]These authors contributed
equally to this work

**Competing interest:** The authors
declare that no competing
interests exist.

**Reviewing Editor:** Kate M
Wassum, University of California,
Los Angeles, United States

## eLife assessment

This study presents **useful** findings for how sevoflurane anesthesia modulates the activity of corticotropin-releasing hormone neurons in the paraventricular nucleus of the hypothalamus and how manipulation of such PVHCRH neurons influences anesthesia and post-anesthesia responses. The technical approaches are **solid** and the data presented is largely clear. Whether PVHCRH neurons are critical for the mechanisms of sevoflurane anesthesia is a direction for the future.

**Abstract** General anesthesia (GA) is an indispensable procedure necessary for safely and compassionately administering a significant number of surgical procedures and invasive diagnostic tests. However, the undesired stress response associated with GA causes delayed recovery and even increased morbidity in the clinic. Here, a core hypothalamic ensemble, corticotropin-releasing hormone neurons in the paraventricular nucleus of the hypothalamus (PVH[CRH] neurons), is discovered to play a role in regulating sevoflurane GA. Chemogenetic activation of these neurons delay the induction of and accelerated emergence from sevoflurane GA, whereas chemogenetic inhibition of PVH[CRH] neurons accelerates induction and delays awakening. Moreover, optogenetic stimulation of PVH[CRH] neurons induce rapid cortical activation during both the steady and deep sevoflurane GA state with burst-suppression oscillations. Interestingly, chemogenetic inhibition of PVH[CRH] neurons relieve the sevoflurane GA-elicited stress response (e.g., excessive self-grooming and elevated corticosterone level). These findings identify PVH[CRH] neurons modulate states of anesthesia in sevoflurane GA, being a part of anesthesia regulatory network of sevoflurane.

## Introduction

In the past century, volatile general anesthetics have gained worldwide popularity in clinics owing to their favorable properties, including a more pleasant odor, higher potency, low toxicity, and rapid recovery (*Robinson and Toledo, 2012*). General anesthesia (GA) is a combination of behavioral and physiological states induced and maintained primarily by pharmacologic agents. As an ideal volatile anesthetic agent, sevoflurane is commonly used for rapid and smooth induction and maintenance for GA (*Brioni et al., 2017*). However, undesirable excessive stress responses of patients toward sevoflurane GA may result in prolonged rehabilitation and long-term prognosis (*Marana et al., 2013*).

There is increasing evidence supporting the 'shared circuits hypothesis' of GA and sleep, in which different GA drugs may exert hypnotic effect through a shared brain network with wake–sleep

regulation (*Lydic, 1996*). The paraventricular nucleus of the hypothalamus (PVH) is one of the pivotal wake-promoting nuclei (*Wang et al., 2022*; *Chen et al., 2021*), which has dense reciprocal connections with numerous neuroanatomical sites (*Chen et al., 2021*) that promote both wakefulness and emergence from inhaled GA, including the lateral septum (LS) (*Wang et al., 2021a*), paraventricular thalamus (PVT) (*Ao et al., 2021*; *Ren et al., 2018*), and parabrachial nucleus (PB) (*Wang et al., 2019*; *Luo et al., 2018b*; *Xu et al., 2021b*; *Kaur et al., 2013*; *Qiu et al., 2016*). The PVH mainly consists of glutamatergic neurons (*Qin et al., 2018*). Specifically activating PVH glutamatergic neurons using chemogenetic approaches potently enhances the proportion of wakefulness, while lesions of PVH glutamatergic neurons have been shown to lead to serious hypersomnia in both mice and patients (*Wang et al., 2022*; *Chen et al., 2021*). As a subgroup of PVH glutamatergic neurons (*Xu et al., 2020b*), corticotropin-releasing hormone neurons in the PVH (PVH[CRH]) neurons are involved in hypothalamic circuitry underlying stress-induced insomnia (*Li et al., 2020*). Mild optogenetic stimulation of PVH[CRH] neurons evoked persistent wakefulness, which mimicked insomnia elicited by restraint stress (*Li et al., 2020*). Collectively, this evidence suggests that PVH[CRH] neurons may play a role in regulating states of consciousness in GA.

The perioperative stress response is a spectrum of physiological changes occurring throughout different systems in the body, including neuroendocrine, metabolic, immunological, and hematological changes (*Iwasaki et al., 2015*), as well as behavioral changes (*Jöhr, 2002*). A prolonged stress response may stimulate excess cortisol release, immunosuppression, and an increase in proteolysis, which delays postoperative recovery and even leads to increased morbidity (*Huiku et al., 2007*; *Desborough, 2000*). As an important contributing factor of perioperative stress, GA-related stress has been neglected. Accumulated evidence has shown that the hypothalamic–pituitary–adrenal (HPA) axis is actively involved in the surgical stress during and after anesthesia (*Besnier et al., 2017*). Postoperatively elevated cortisol secretion is considered a part of the surgical stress response (*Gögenur et al., 2007*). Distinct from other hypnotic agents, sevoflurane influences the endocrine response by increasing adrenocorticotrophic hormone and cortisol levels (*Mizutani et al., 1998*; *Kostopanagiotou et al., 2010*). Notably, PVH[CRH] neurons serve as the initial node of the HPA axis, transducing the neuronal stress signal into a glucocorticoid secretion signal (*Ramot et al., 2017*). Therefore, we hypothesize that PVH[CRH] neurons may also act as a crucial node modulating the stress response of sevoflurane GA.

To test our hypothesis, we first performed global screening of active neurons during the post-anesthesia period in the mouse brain. We found that PVH[CRH] neurons were preferentially active during the post-anesthesia period. After observing changes in calcium signals in PVH[CRH] neurons during the induction and maintenance of, and recovery from sevoflurane GA along with a robust post-anesthesia stress response, we used chemogenetic and optogenetic manipulations to demonstrate how PVH[CRH] neurons modulate sevoflurane GA. The results showed that the activation of PVH[CRH] neurons conferred resistance to sevoflurane GA. Strikingly, inhibition of PVH[CRH] neurons facilitated induction of sevoflurane GA and alleviated post-anesthesia stress responses. Hence, our findings uncover a vital brain site that modulates the state of consciousness as well as the stress response in sevoflurane GA.

## Results
### Identification of activated neurons during the post-anesthesia period

Initially, to gain access to the active neurons during the post-anesthesia period, we subjected the mice to either sevoflurane-oxygen anesthesia (sevo) or oxygen exposure alone (control) for 30 min, followed by examining the whole-brain c-fos expression 3 hr later (*Figure 1A*). Considering the time of the post-anesthesia stress response and the peak time of c-fos protein expression (2–3 hr) after neuronal activation (*Xiu et al., 2014*), a higher level of immunoreactivity to c-fos was observed in the PVH of sevo mice than that in control mice (*Figure 1B, C*). Increased immunoreactivity was also observed in the anterior olfactory nucleus, LS, ventromedial preoptic nucleus, locus coeruleus, and nucleus of the solitary tract (Sol) (*Figure 1B, C* and *Figure 1—figure supplement 1*). Interestingly, the PVH, traditionally considered a classic stress center (*Godoy et al., 2018*), was the brain region with the highest relative expression in the sevo group. In contrast, other stress-related brain regions, such as the nucleus accumbens, the ventral tegmental area, and the central amygdaloid nucleus, did not exhibit robust c-fos immunoreactivity (*Figure 1—figure supplement 2A–D*). Additionally,

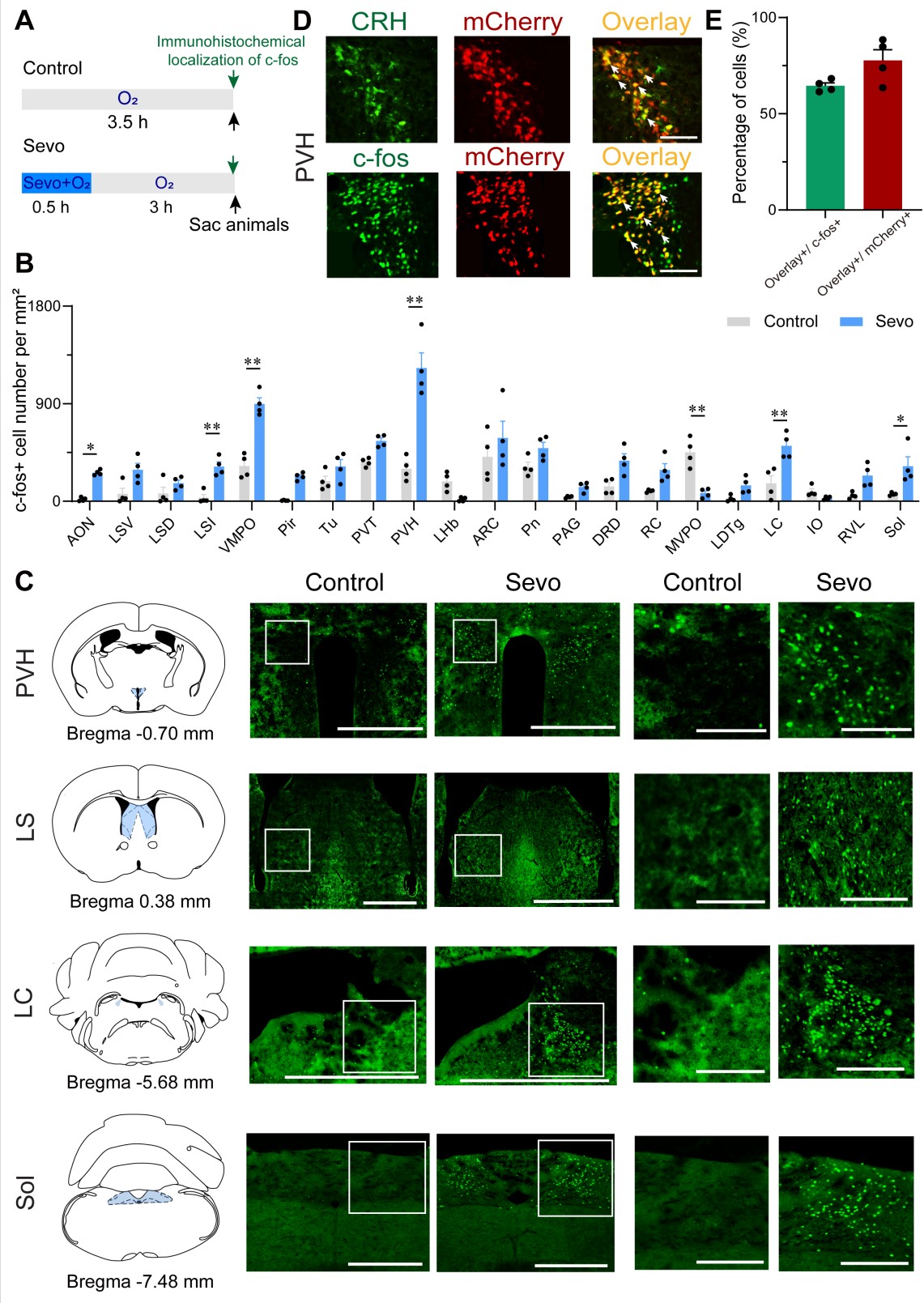

**Figure 1.** Whole-brain mapping of activated neurons during the post-anesthesia period. (**A**) Experimental timeline for c-fos visualization. Animals were exposed to sevoflurane or oxygen for 30 min and were sacrificed 3 hr after the treatment. (**B**) Quantification of the number of c-fos+ cells per mm$^2$ in different brain regions ($n$ = 4, unpaired two-tailed $t$-test, *p < 0.05, **p < 0.01). AON, anterior olfactory nucleus; ARC, arcuate hypothalamic nucleus; DRD, dorsal raphe nucleus, dorsal part; IO, inferior olive; LC, locus coeruleus; LDTg, laterodorsal tegmental nucleus; LHb, lateral habenular nucleus;

*Figure 1 continued on next page*

*Figure 1 continued*

LSD, lateral septal nucleus, dorsal part; LSI, lateral septal nucleus, intermediate part; LSV, lateral septal nucleus, ventral part; MVPO, medioventral periolivary nucleus; PAG, periaqueductal gray; Pir, piriform cortex; Pn, pontine nuclei; PVH, paraventricular nucleus of the hypothalamus; PVT, paraventricular thalamus; RC, raphe cap; RVL, rostroventrolateral reticular nucleus; Sol, nucleus of the solitary tract; Tu, olfactory tubercle; VMPO, ventromedial preoptic nucleus. (**C**) Representative images of c-fos immunoreactivity in the PVH, LS, LC, and Sol for a control and an experimental animal exposed to sevoflurane general anesthesia. Scale bar, 500 µm (left), 250 µm (right, enlarged). (**D**) Representative images showing colocalization of CRH immunoreactivity and mCherry expression (upper panel, scale bar, 100 µm), c-fos immunoreactivity and mCherry expression (bottom panel, scale bar, 100 µm) in the PVH of CRH-Cre mice. Arrowheads indicate co-labeled neurons. (**E**) Quantification of the percentage of mCherry+ cells in the c-fos+ population (green) and the percentage of c-fos+ cells in the mCherry+ population (red) in the PVH.

The online version of this article includes the following source data and figure supplement(s) for figure 1:

**Figure supplement 1.** Whole-brain mapping of active neurons during the post-anesthesia period.

**Figure supplement 2.** Representative images of brain regions without robust c-fos immunoreactivity.

**Figure supplement 3.** Validation of CRH-Cre mice and CRH antibody.

**Figure supplement 3—source data 1.** Original file of the full raw uncropped, unedited gels of RT-PCR for the validation of CRH-Cre mice.

**Figure supplement 3—source data 2.** Figure with the uncropped gels with the relevant bands clearly labeled of RT-PCR for the validation of CRH-Cre mice.

the medioventral periolivary nucleus showed decreased c-fos levels after inhalation of sevoflurane (*Figure 1B* and *Figure 1—figure supplement 1C*). To confirm specific markers for the c-fos-positive neurons in the PVH, we labeled PVH^CRH neurons by injecting a Cre-dependent adeno-associated virus (AAV) expressing mCherry into the PVH of CRH-Cre mice (*Figure 1—figure supplement 3*, *Figure 1—figure supplement 3—source data 1 and 2*), and found that mCherry-expressing neurons showed high proportion of CRH immunoreactivity in the PVH (*Figure 1D*, *top panel*). As shown by the colocalization of c-fos- and CRH-immunoreactive neurons, approximately 65.71% of activated PVH neurons were CRH neurons, and approximately 73.91% of CRH neurons in the PVH were activated after exposure to sevoflurane GA (*Figure 1D*, *bottom panel*, *Figure 1E*). These results indicate that sevoflurane GA elicits robust activation of PVH^CRH neurons.

## Population activity of PVH^CRH neurons in response to sevoflurane GA

To reveal the real-time responses of PVH^CRH neurons to sevoflurane GA, we monitored the temporal dynamics of PVH^CRH neuronal activity in CRH-Cre mice during the whole process of sevoflurane GA (i.e., induction, maintenance, emergence, and recovery phases) using in vivo fiber photometry. A Cre-dependent AAV expressing the fluorescent calcium indicator, jGCaMP7b (AAV2/9-hSyn-DIO-jGCaMP7b-WPRE-pA), was injected into the PVH of CRH-Cre mice (*Figure 2A*). The expression of the virus is shown in *Figure 2B*. During the whole experimental process, the mice were placed in a vertical cylindrical cage for sevoflurane GA (*Figure 2C*). As shown in *Figure 2D*, the neural activity declined with exposure to sevoflurane and increased with the cessation of sevoflurane inhalation. In detail, compared to the awake baseline, PVH^CRH neurons showed significantly decreased neuronal activity during the 30 min exposure to 1.6% sevoflurane. By analyzing the $Ca^{2+}$ signals over four periods (*Figure 2E*, *Figure 2—source data 1*): pre (−30 to 0 min), during (anesthesia period, 0 to 30 min), post 1 (30 min to time to recovery of righting reflex [RORR]), and post 2 (RORR to baseline), we found that the population activities of PVH^CRH neurons were significantly blunted by sevoflurane exposure (during vs. pre: −31.11% ± 9.11%; p = 0.027) and gradually returned during the recovery period (post 1 vs. during: 5.84% ± 1.19%; p = 0.651). These findings suggest that the activities of PVH^CRH neurons were altered across distinct concentrations of sevoflurane.

Notably, PVH^CRH neurons exhibited higher $Ca^{2+}$ signals compared to the awake baseline after a 30-min cessation of sevoflurane inhalation. After comparing the $Ca^{2+}$ signals of the awake baseline period (pre) and post-firing period (post 2), we found that the population activities of PVH^CRH neurons increased potently during post 2 than baseline period (*Figure 2D-E*, pre vs. post 2: 40.38% ± 4.75%; p = 0.002, *Figure 2—source data 1*). Interestingly, this hyperactivity phenomenon, lasting approximately 1 hr before returning to baseline levels, was accompanied by enhanced self-grooming (*Figure 2F*, *Figure 2—source data 1*), which is considered a stress-related behavior (*Xu et al., 2019*). These results prompted us to speculate that the hyperactivities of PVH^CRH neurons may play a vital role in the post-anesthesia stress response.

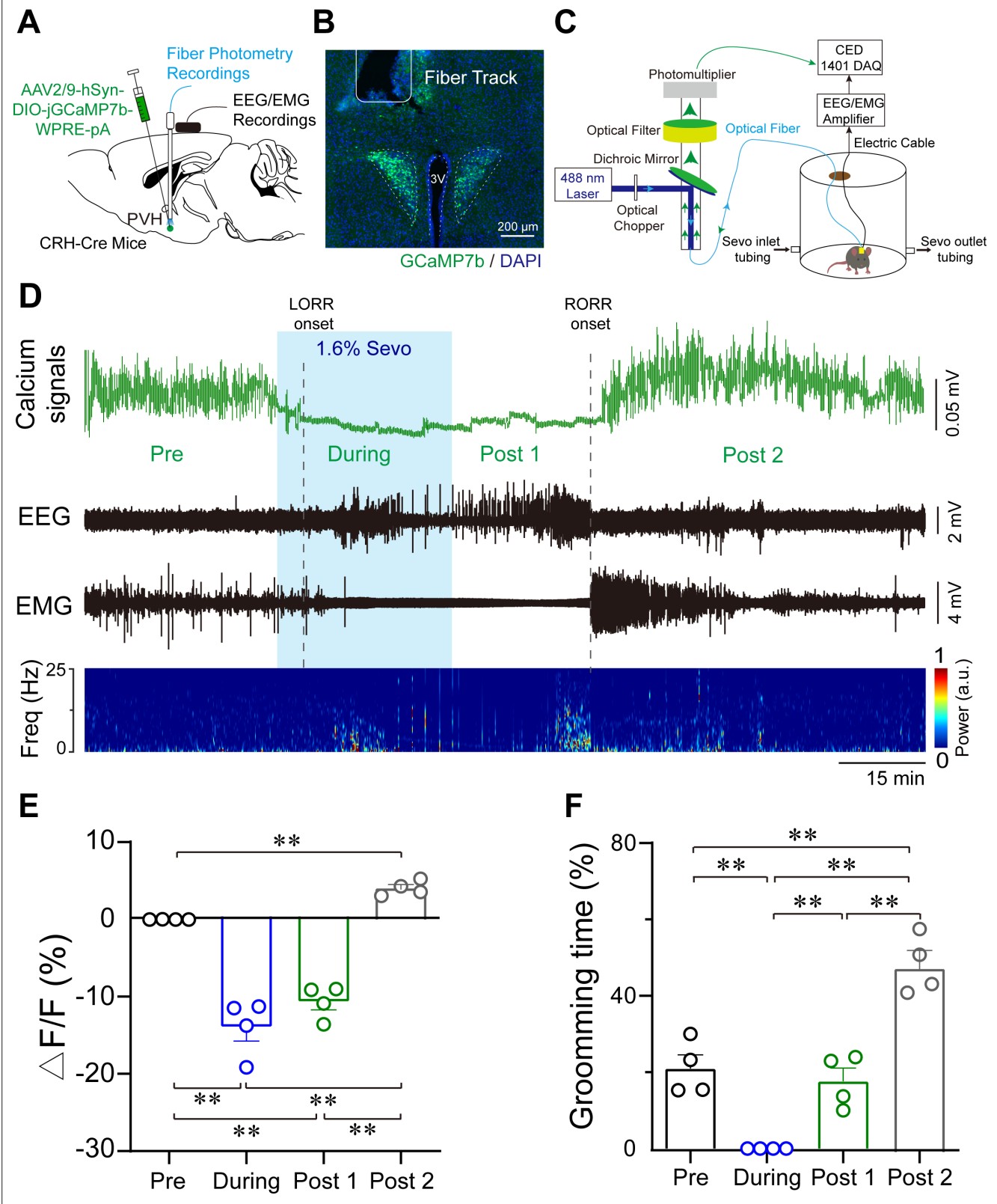

**Figure 2.** Population activities of PVH[CRH] neurons in response to sevoflurane general anesthesia. (**A**) Diagram of the virus injection, electroencephalogram (EEG)/electromyography (EMG) electrode, and optic fiber implantation sites of CRH-Cre mice. (**B**) jGCaMP7b/DAPI immunofluorescence in CRH neurons and track of the optic fiber implanted above the paraventricular nucleus of the hypothalamus (PVH); scale bar, 200 μm. Viral expression of jGCaMP7b and placement of the fiber-optic probe above the PVH. (**C**) Schematic of the recording configuration. (**D—E**)

*Figure 2 continued on next page*

*Figure 2 continued*

Time courses of Ca$^{2+}$ signals (**D**) and quantification of Ca$^{2+}$ signal changes before, during, and after (post 1 and post 2 periods) sevoflurane inhalation (**E**), $n$ = 4, $F_{(3, 12)}$ = 61.49, $p < 0.001$, one-way ANOVA with Tukey's post hoc test; pre vs. during, pre vs. post 1, during vs. post 2, post 1 vs. post 2, **$p < 0.001$; pre vs. post 2, $p$ = 0.0894; during vs. post 1, $p$ = 0.2012. Freq, frequency; LORR, loss of right reflexing; RORR, recovery of right reflexing; Sevo, sevoflurane. (**F**) Time percentage of self-grooming before, during, and after (post 1 and post 2 periods) sevoflurane inhalation ($n$ = 4, $F_{(3, 12)}$ = 76.87, $p < 0.001$, one-way repeated measures ANOVA with Tukey's post hoc test; pre vs. during, pre vs. post 1, $p$ = 0.0005; pre vs. post 2 during vs. post 2, post 1 vs. post 2, **$p < 0.0001$; during vs. post 1, $p > 0.9999$).

The online version of this article includes the following source data for figure 2:

**Source data 1.** Original data for analysis displayed in *Figure 2E,F*.

## Characterizations of sevoflurane GA-induced grooming

Stress in animals often results in grooming and other repetitive behaviors (*Troisi, 2002*). As these displacement activities are believed to have adaptive values to stress (*Song et al., 2016*; *Kalueff et al., 2016*), we employed excessive self-grooming as an indicator of stress response of sevoflurane GA. Next, we interrogated the characteristics of sevoflurane GA-induced grooming.

Considering that grooming behaviors are context-sensitive and could be reflected in their sequence patterns or microstructure, we analyzed these characteristics of sevoflurane GA-induced grooming. Following the termination of sevoflurane GA, mice stayed awake for approximately 1.5 hr, during which they exhibited robust self-grooming with increased total time spent grooming and duration per bout (*Figure 3—figure supplement 1*, *Figure 3—figure supplement 1—source data 1*). Because of the consistent trend, we divided the post-anesthesia period (post 2) into two sections (0–50 vs. 50–100 min). Compared to the former period, the latter period showed higher percentages of incorrect phase transition and interrupted bouts (*Figure 3A, B*, *Figure 3—source data 1*, see Materials and methods), indicating elevated stress levels; thus, we selected the latter as the representative period to investigate sevoflurane GA-induced grooming.

Subsequently, we compared sevoflurane GA-induced grooming with four grooming models related to physical and emotional stress, including those elicited by free swimming, water spray, physical attack, and body restraint (*Figure 3C*). The former two models induce more physical stress by moistening the fur, while the latter two models are often considered models for emotional stress (*Dayas et al., 2001*). In line with previous reports (*Song et al., 2016*; *Kalueff et al., 2016*; *Kalueff et al., 2007*; *Kalueff and Tuohimaa, 2005*), the mice in these two emotional stress-related groups spent a higher percentage of time paw licking, while those in the two fur moistening groups spent most time in grooming the body (*Figure 3—figure supplement 1*, *Figure 3—figure supplement 1—source data 1*). Interestingly, for transition bouts and grooming bouts per min, sevoflurane GA-induced grooming during 50–100 min was close to spontaneous grooming (*Figure 3E–F*, *Figure 3—source data 1*), but it shared no similarities with these four grooming models in terms of the mean bout duration (*Figure 3D, G*, *Figure 3—source data 1*). A plot of the number of times spent grooming different body regions also showed no similarity among sevoflurane GA-induced grooming and other models (*Figure 3H* and *Figure 3—figure supplement 1*, *Figure 3—figure supplement 1—source data 1*). The combination of these parameters suggested a low level of stress at the beginning of the post-anesthesia period followed by a higher and unique level of stress with different parameters (longest mean bout duration, lowest bout frequency, and average distribution of grooming time spent grooming different body parts).

## Modulation of PVH$^{CRH}$ neurons altered the induction and emergence of sevoflurane GA

To investigate the role of PVH$^{CRH}$ neurons during sevoflurane GA, we micro-injected AAV-DIO-hM3Dq-mCherry or AAV-DIO-hM4Di-mCherry or AAV-DIO-mCherry into the bilateral PVH of CRH-Cre mice, respectively (*Figure 4A*). The c-fos immunoreactivity overlaid with hM3Dq or hM4Di is shown in *Figure 4B*. We then performed behavioral experiments to determine the effect of chemogenetic activation or inhibition of PVH$^{CRH}$ neurons on the sensitivity, induction, and emergence of sevoflurane GA. First, the sensitivity of sevoflurane GA was evaluated by a minimum inhaled sevoflurane concentration that evoked loss of righting reflex (LORR), which is a potent behavioral indicator of the onset of GA. We found that PVH$^{CRH}$ neuron-activated mice were resistant to higher concentrations of sevoflurane

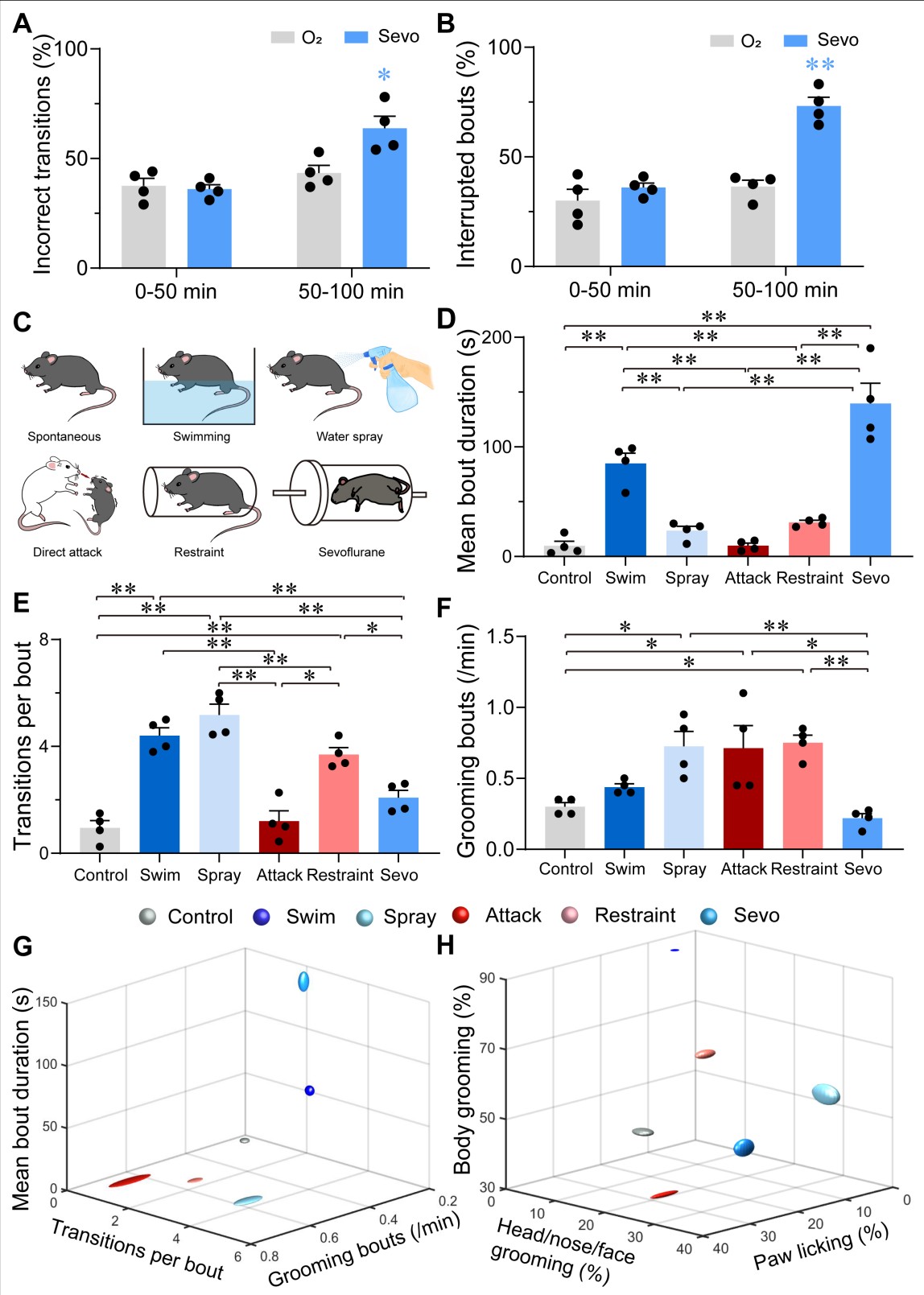

**Figure 3.** Characterizations of sevoflurane general anesthesia-induced grooming. (**A**) Quantification of incorrect transitions with respect to the cephalocaudal sequence of stereotypic grooming patterns in the post-anesthesia period ($n$ = 4, two-way ANOVAs followed by Sidak's test, $F_{(1, 6)}$ = 8.646, p = 0.0259, 0–50 min: $O_2$ vs. sevo, p = 0.9545, $t$ = 0.2767, df = 12, 95% CI = −12.34 to 15.34; 50–100 min, $O_2$ vs. sevo, p = 0.0055, $t$ = 3.753, df = 12, 95% CI = −34.19 to −6.504). (**B**) Number of interrupted bouts in the post-anesthesia period ($n$ = 4, two-way ANOVAs followed by Sidak's test, $F_{(1, 6)}$

*Figure 3 continued on next page*

*Figure 3 continued*

= 19.14, p = 0.0047, 0–50 min: O$_2$ vs. sevo, p = 0.4770, *t* = 1.139, df = 12, 95% CI = −19.45 to 7.446; 50–100 min, O$_2$ vs. sevo, **p < 0.01, *t* = 6.969, df = 12, 95% CI = −50.15 to −23.26). (**C**) Six representative grooming models, including spontaneous (control), swimming, water spray, physical attack, body restraint, and sevoflurane GA-induced grooming. (**D-F**) The mean bout duration (**D**, $F_{(5, 18)}$ = 34.52, p < 0.0001), transitions per bout (**E**, $F_{(5, 18)}$ = 30.32, p < 0.0001), and grooming frequency (bouts per min, **F**, $F_{(5, 18)}$ = 7.935, p = 0.0004) varied across the models (*n* = 4, one-way ANOVA with Tukey's post hoc test. *p < 0.05; **p < 0.01). (**G-H**) 3D plot of bout frequency, bout duration, and transitions per bout (**G**); the percentage of time spent grooming different body parts (**H**). The dimension of the symbol along an axis is defined by the SD of the corresponding parameter.

The online version of this article includes the following source data and figure supplement(s) for figure 3:

**Source data 1.** Original data for analysis displayed in *Figure 3*.

**Figure supplement 1.** Characterization of sevoflurane general anesthesia -induced grooming.

**Figure supplement 1—source data 1.** Original data for analysis displayed in *Figure 3—figure supplement 1*.

than the vehicle group (*Figure 4D*, *Figure 4—source data 1*), whereas the PVH$^{CRH}$ neuron-inhibited mice were more vulnerable to sevoflurane GA (*Figure 4E*, *Figure 4—source data 1*). The right-shift of the dose–response curve along with the augmentation of the 50% effective concentration (EC$_{50}$) in the PVH$^{CRH}$ neuron-activated mice suggested that PVH$^{CRH}$ neuron activation lowered sevoflurane sensitivity during sevoflurane induction. Conversely, the left-shift of the dose–response curve combined with the decrease in EC$_{50}$ in the PVH$^{CRH}$ neuron-inhibited mice revealed higher sevoflurane sensitivity during induction. According to the dose–response curves (*Figure 4—source data 2*), the EC$_{50}$ of sevoflurane in term of LORR from the PVH$^{CRH}$ neuron-activated mice (*n* = 10) was 1.650% vs. 1.516% of the vehicle group, and the EC$_{50}$ of sevoflurane in term of RORR also increased (*Figure 4—figure supplement 1A*, *Figure 4—figure supplement 1—source data 1*). As for the chemogenetic inhibition experiments, the EC$_{50}$ of the PVH$^{CRH}$ neuron-inhibited mice (*n* = 8) was 1.337% vs. 1.500% of the vehicle mice, and the EC$_{50}$ of sevoflurane in term of RORR also reduced (*Figure 4—figure supplement 1B*, *Figure 4—figure supplement 1—source data 1*). The EC$_{50}$ of sevoflurane in term of either RORR or LORR did not show significant change in control group (*Figure 4—figure supplement 1C-D*, *Figure 4—figure supplement 1—source data 1*).

We then measured the induction time (time to LORR) and emergence time (RORR) following the experiment procedure outlined in *Figure 4C*. In the chemogenetic activation experiments, clozapine-*N*-oxide (CNO) pre-treatment led to increased induction time after 2% sevoflurane exposure (*Figure 4F*, *Figure 4—source data 1*). Concomitantly, the PVH$^{CRH}$ neuron-activated mice demonstrated faster emergence from 2% sevoflurane GA (*Figure 4G*, *Figure 4—source data 1*). As for the chemogenetic inhibition manipulation, an obvious shortened induction time and a prolonged emergence time were observed after CNO pre-treatment (*Figure 4F,G*, *Figure 4—source data 1*). These results further imply that chemogenetically activating PVH$^{CRH}$ neurons increased resilience to sevoflurane and promoted awakening from sevoflurane GA, whereas chemogenetic inhibition manipulation of PVH$^{CRH}$ neurons exerted contrary effects. To further examine the role of PVH$^{CRH}$ neurons in sevoflurane GA, we genetic ablated PVH$^{CRH}$ neurons (*Figure 4—figure supplement 2*) and found similar effects to the inhibition group in terms of dose–response curve. The EC$_{50}$ of sevoflurane in terms of LORR decreased from 1.491% to 1.358% compared with control group (*Figure 4—figure supplement 2*, *Figure 4—figure supplement 2—source data 1*), and lesion of PVH$^{CRH}$ neurons decreased EC$_{50}$ of sevoflurane on RORR from 1.420% to 1.326% (*Figure 4—figure supplement 2*, *Figure 4—figure supplement 2—source data 1*). Meanwhile, genetic ablation of PVH$^{CRH}$ neurons significantly facilitated induction and slowed emergence of sevoflurane GA (*Figure 4—figure supplement 2*, *Figure 4—figure supplement 2—source data 1*), which showed lower induction time and longer emergence time than the inhibition group.

Considering the temporal limitation of chemogenetic manipulation, we next employed optogenetic methods with millisecond-scale control of neuronal activities to verify the role of PVH$^{CRH}$ neurons in controlling sevoflurane GA. To this end, we stereotaxically injected AAVs expressing channelrhodopsin-2 or only mCherry into the bilateral PVH of CRH-Cre mice with optical fibers implanted (*Figure 4—figure supplement 3A*). We verified the expression of ChR2 and the optical fiber locations after behavior experiments (*Figure 4—figure supplement 3B*). Mice were placed in the same vertical cylindrical cage as the apparatus for in vivo fiber photometry for sevoflurane GA. Optical blue-light stimulation (5 ms, 30 Hz, 20–30 mW/mm²) was applied at the onset of induction and ended until LORR or started at the end of sevoflurane inhalation and continued until RORR (*Figure 4—figure*

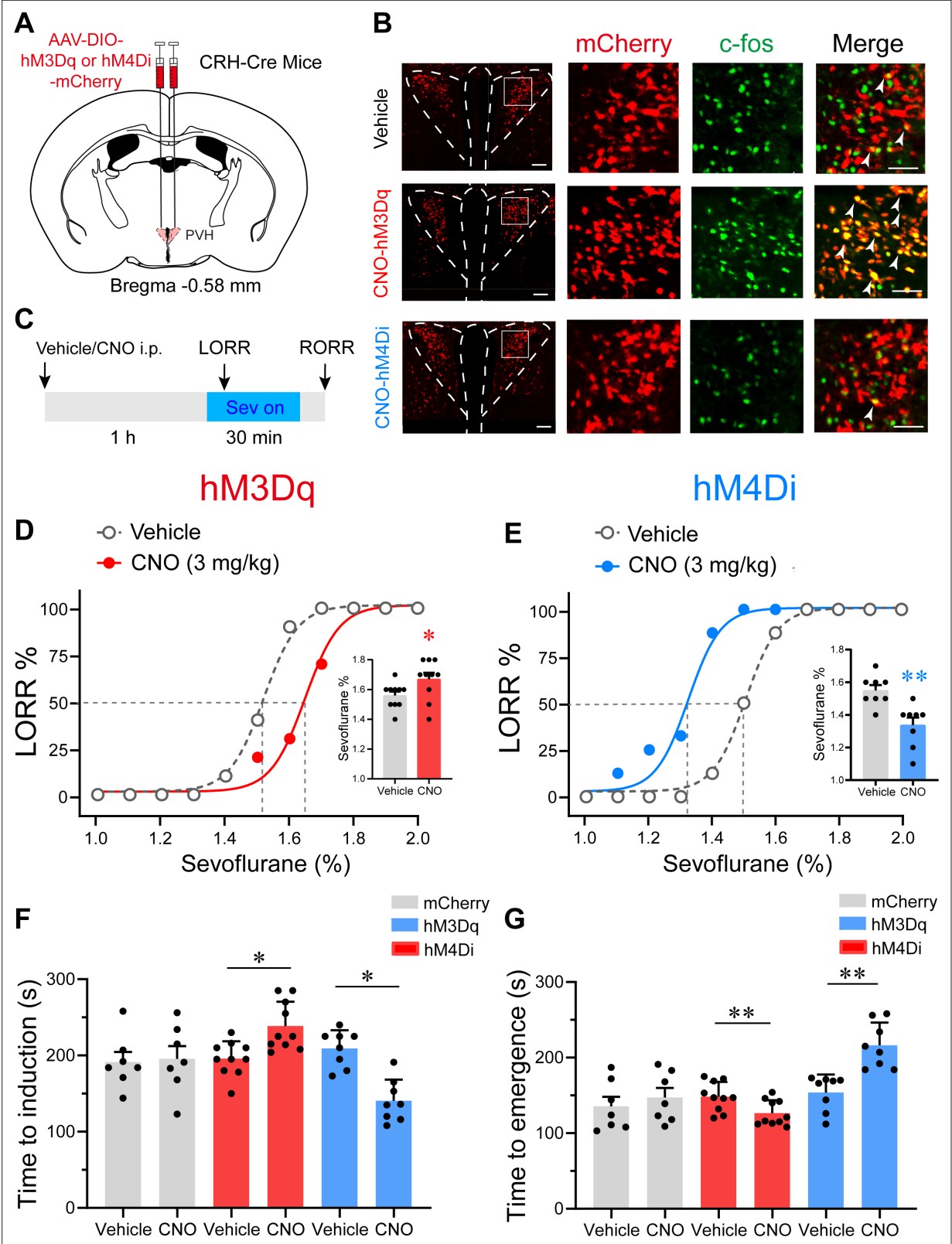

**Figure 4.** Chemogenetic modulation of PVH^CRH neurons bidirectionally altered induction of and emergence from sevoflurane general anesthesia. (**A**) Schematic of AAV-DIO-hM3Dq-mCherry or AAV-DIO-hM4Di-mCherry or AAV-DIO-mCherry injected into the paraventricular nucleus of the hypothalamus (PVH) of CRH-Cre mice. (**B**) Left: representative images of mCherry/c-fos immunofluorescence in CRH neurons after vehicle or clozapine-*N*-oxide (CNO) treatment; scale bars, 200 μm. Right: magnified images are shown; scale bar, 200 μm. Arrowheads indicate co-labeled neurons. (**C**)

*Figure 4 continued on next page*

*Figure 4 continued*

Timelines of sevoflurane anesthesia-related behavioral tests measuring induction time (loss of righting reflex, LORR) and emergence time (recovery of righting reflex, RORR). (**D, E**) Dose–response curves showing the percentages of mice exhibiting LORR in response to incremental sevoflurane concentrations for the vehicle and CNO groups. Inset: the sevoflurane concentrations at which each mouse exhibited LORR are shown (**D**, hM3Dq group, $n = 10$, paired $t$-test, $p = 0.0411$, $t = 4.714$, df = 9, 95% CI = 0.05 to 0.16; **E**, hM4Di group, $n = 8$, paired $t$-test, $p = 0.0021$, $t = 9.375$, df = 7, 95% CI = −0.266 to −0.1589). (**F**) Induction time with 2% sevoflurane exposure after intraperitoneal injections of vehicle or CNO for 1 hr (mCherry group, $n = 7$, paired $t$-test, $p = 0.847$, $t = 0.2014$, df = 6, 95% CI = −47.77 to 56.35; hM3Dq group, $n = 10$, paired $t$-test, $p = 0.0498$, $t = 5.545$, df = 7, 95% CI = 32.26 to 80.24; hM4Di group, $n = 8$, paired $t$-test, $p = 0.0060$, $t = 4.573$, df = 7, 95% CI = −104.1 to −33.14). (**G**) Emergence time with 2% sevoflurane exposure for 1 hr after intraperitoneal injections of vehicle or CNO (mCherry group, $n = 7$, paired $t$-test, $p = 0.8298$, $t = 0.9286$, df = 6, 95% CI = −19.15 to 42.58; hM3Dq group, $n = 10$, paired $t$-test, $p = 0.0048$, $t = 4.057$, df = 7, 95% CI = −43.72 to −11.53; hM4Di group, $n = 8$, paired $t$-test, $p = 0.0057$, $t = 3.922$, df = 7, 95% CI = 24.77 to 99.98). *$p < 0.05$, **$p < 0.01$.

The online version of this article includes the following source data and figure supplement(s) for figure 4:

**Source data 1.** Original data for analysis displayed in *Figure 4D-G*.

**Source data 2.** Mean value of EC$_{50}$ for sevoflurane dose–response curves.

**Figure supplement 1.** Dose–response curves showing the percentages of mice exhibiting LORR (**C**) or recovery of righting reflex (RORR) (**A, B, D**) in response to incremental or decreased sevoflurane concentrations for the PVH$^{CRH}$ neurons activation (**A**), inhibition (**B**), and control (**C, D**) groups.

**Figure supplement 1—source data 1.** Original data for analysis displayed in *Figure 4—figure supplement 1*.

**Figure supplement 2.** Lesion of PVH$^{CRH}$ neurons facilitated induction of and delayed emergence from sevoflurane general anesthesia.

**Figure supplement 2—source data 1.** Original data for analysis displayed in *Figure 4—figure supplement 2C-F*.

**Figure supplement 3.** Optogenetic stimulation of PVH$^{CRH}$ neurons delayed induction and facilitated emergence from sevoflurane general anesthesia.

**Figure supplement 3—source data 1.** Original data for analysis displayed in *Figure 4—figure supplement 3D-E*.

supplement 3C*). We found that optical activation of PVH$^{CRH}$ neurons prominently delayed the induction process (*Figure 4—figure supplement 3D*, *Figure 4—figure supplement 3—source data 1*) and accelerated the recovery process (*Figure 4—figure supplement 3E*, *Figure 4—figure supplement 3—source data 1*) compared to the mCherry control, indicating that optical activation of PVH$^{CRH}$ neurons could prompt the emergence from sevoflurane GA.

## Optogenetic stimulation of PVH$^{CRH}$ neurons induced rapid emergence from steady sevoflurane GA state

We next applied 60 s of optical blue-light stimulation (5 ms, 30 Hz, 20–30 mW/mm$^2$) after mice had maintained LORR for 30 min during steady GA state maintained by 2% sevoflurane. At this stage, the electroencephalogram (EEG) patterns of the mice were characterized by an increase in low-frequency, high-amplitude activity, which resembles the EEG pattern of non-rapid eye movement sleep (slow–wave sleep). The optogenetics experiments were performed in the similar apparatus with partially sealed chamber in the *Figure 2C*. Photostimulation of PVH$^{CRH}$ neurons at 30 Hz during this state reliably elicited a brain-state transition from slow-wave activity to a low-amplitude, high-frequency activity in ChR2 mice, but not in mCherry mice (*Figure 5A, B*). We also observed behavioral emergence (including body movements of limbs, head, and tail, righting, and walking) (*Figure 5—source data 1*) combined with enhanced electromyography (EMG) activity after photostimulation at 30 Hz. Spectral analysis of EEG data revealed that acute photostimulation of PVH$^{CRH}$ neurons at 30 Hz elicited a prominent decrease in delta power (60.83% ± 1.35% vs. 40.23% ± 5.30%; $p = 0.045$, $t = 7.406$, df = 40, 95% CI=12.46 to 26) and an increase in alpha power (7.47% ± 0.41% vs. 12.42% ± 0.93%; $p = 0.033$, $t = 1.721$, df = 40, 95% CI = −11.24 to 2.303) and beta power (6.01% ± 0.245% vs. 11.23% ± 1.27%; $p = 0.025$, $t = 1.727$, df = 40, 95% CI = −11.25 to 2.286; *Figure 5D, F*, *Figure 5—source data 2*). There was no significant difference observed in the EEG spectrum of pre-stimulation and stimulation periods in mCherry mice (*Figure 5C, E*, *Figure 5—source data 2*). These findings revealed that optogenetic activation of PVH$^{CRH}$ neurons was sufficient to induce cortical activation and behavioral emergence during the steady sevoflurane GA state.

## Optogenetic stimulation of PVH$^{CRH}$ neurons induced EEG activation during burst-suppression oscillations induced by deep sevoflurane GA

We further explored the role of PVH$^{CRH}$ neurons in the regulation of deep sevoflurane GA, during the burst-suppression phase, in which the EEG pattern is characterized by flat periods interspersed with

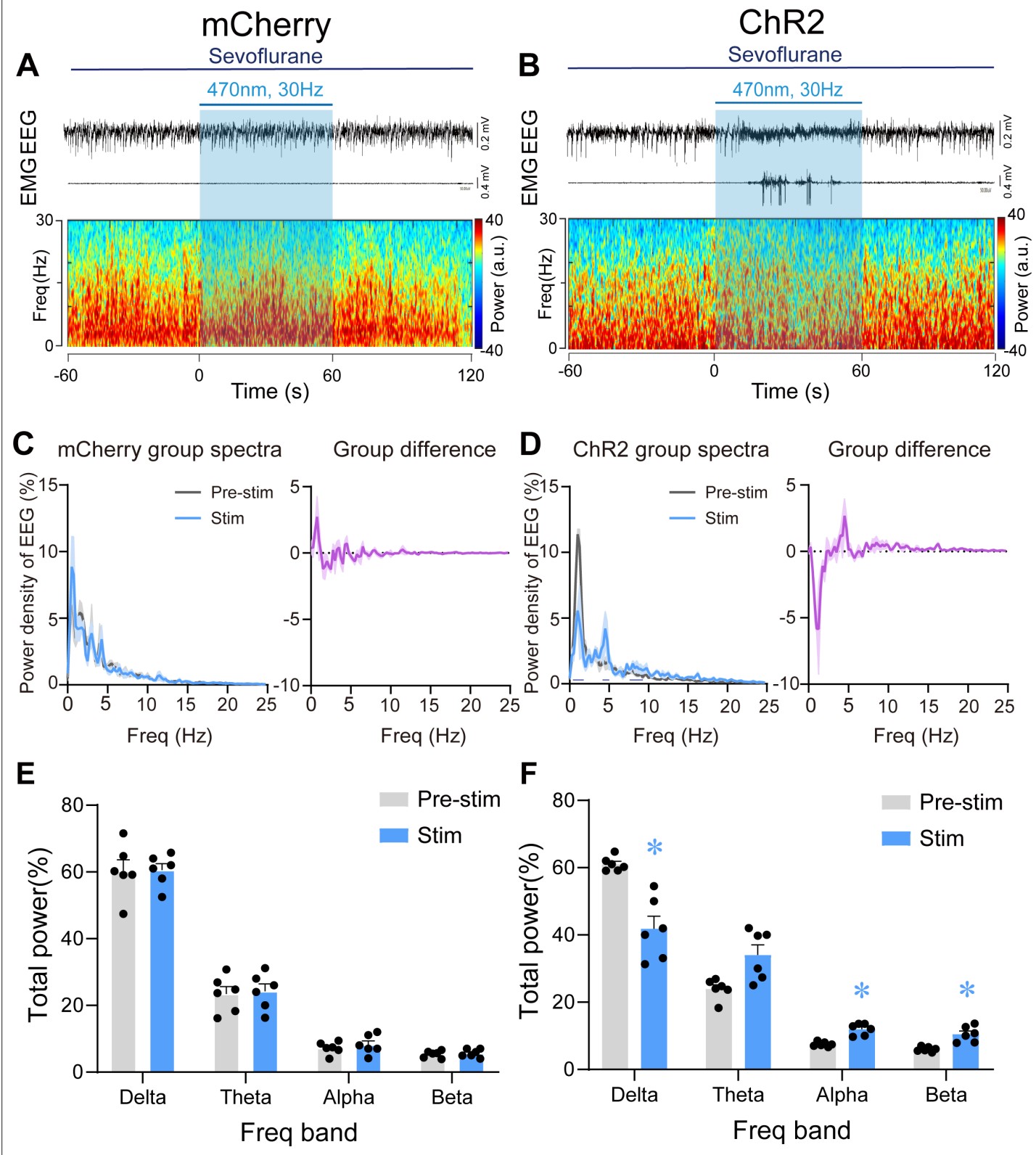

**Figure 5.** Optogenetic stimulation of PVH^CRH neurons induced cortical activation and behavioral emergence during continuous steady-state sevoflurane general anesthesia. Typical examples of electroencephalogram (EEG), electromyography (EMG), and EEG power spectral in a mouse injected with AAV-DIO-mCherry (**A**) or AAV-DIO-ChR2-mCherry (**B**) following acute photostimulation (30 Hz, 5 ms, 60 s) during continuous steady-state sevoflurane GA. Time 0 indicates the beginning of photostimulation. The blue shadow indicates the 60 s duration of blue light stimulation. Left: normalized group power

Figure 5 continued

spectral densities from PVH[CRH]-mCherry (**C**) or PVH[CRH]-ChR2 (**D**) mice with pre photostimulation (gray) and photostimulation (blue). Dark blue lines in D indicate the power band with significant difference. Right: differences between normalized group power spectral densities from PVH[CRH]-mCherry (**C**) or PVH[CRH]-ChR2 (**D**). Power percentage changes in cortical EEG before (gray) and during (blue) photostimulation at 30 Hz in PVH[CRH]-mCherry (**E**) or PVH[CRH]-ChR2 mice (**F**) during continuous steady-state sevoflurane GA (n = 6, two-way ANOVAs followed by Sidak's test, *p < 0.05).

The online version of this article includes the following source data for figure 5:

**Source data 1.** Behavioral responses of CRH-Cre mice under sevoflurane steady-state general anesthesia during photostimulation.

**Source data 2.** Original data for analysis displayed in *Figure 5E-F*.

periods of alpha and beta activity (*Bao et al., 2021*). After the maintenance of burst-suppression EEG mode for at least 5 min, we delivered 60 s of optical stimulation (5 ms, 30 Hz, 20–30 mW/mm$^2$) and found that EEG activation mainly occurred with an average latency of 20 s and maintained for a few minutes before returning to burst-suppression mode. No significant difference of EMG activity between pre-/post-stimulation periods of mCherry mice was observed (*Figure 6A* and *Figure 6—figure supplement 1A*) and ChR2 mice (*Figure 6B* and *Figure 6—figure supplement 1B*). Statistical analysis of 1 min of EEG recordings (optical stimulation for 20 and 20 s before and after photostimulation) demonstrated a robust decline in delta power (from 53.85% ± 2.58% to 41.46% ± 2.37%, p = 0.005, df = 60, 95% CI = 12.3 to 22.34) and an increase in theta power (from 24.25% ± 3.50 to 32.07% ± 2.12%, p = 0.017, df = 60, 95% CI = −15.74 to −5.699), alpha power (from 6.72% ± 1.16% to 11.15% ± 1.37%, p = 0.0062, df = 60, 95% CI = −11.69 to −1.652), and beta power (from 4.00% ± 0.55% to 7.56% ± 0.01%, p = 0.0317, df = 60, 95% CI = −10.43 to −0.394; *Figure 6D, G*). As for the mCherry mice, 60 s of blue-light stimulation had no significant effect on either EEG or EMG activity (*Figure 6A, C, E*, *Figure 6—source data 1*). Photostimulation of PVH[CRH] neurons in ChR2 mice also caused a significant reduction in burst suppression ratio (BSR) (Pre-stim vs. Post-stim, p < 0.01, df = 15, 95% CI = 23.53–48.14; Pre-stim vs. Stim, p < 0.01, df = 15, 95% CI = 17.36–41.97; *Figure 6H*, *Figure 6—source data 1*). There were no significant changes in mCherry mice with 30 Hz photostimulation (*Figure 6F*, *Figure 6—source data 1*). These results collectively suggested that PVH[CRH] neuronal activation was sufficient to facilitate EEG cortical activation but not to promote behavioral emergence during deep states of sevoflurane GA with burst-suppression oscillations.

## Chemogenetic inhibition of PVH[CRH] neurons compromised the stress response after sevoflurane GA

Given the hyperactivity of PVH[CRH] neurons and accompanying increase in self-grooming after exposure to sevoflurane GA, we investigated whether PVH[CRH] neurons are recruited while mice are subjected to post-anesthesia stress. We assessed the behavioral effect of chemogenetic inactivation of these neurons via the bilateral expression of the inhibitory hM4Di receptors (*Figure 7A*). Following the protocol shown in *Figure 7B*, we found that inhalation of 1.6% sevoflurane induced higher immunoreactivity to c-fos in the PVH, which was significantly suppressed by chemogenetic inhibition of PVH[CRH] neurons (*Figure 7C*, *Figure 7—source data 1*). Hypothalamic CRH neurons control the circulating levels of corticosteroid stress hormones in the body (*Bittar et al., 2019*). In line with the c-fos expression, serum CRH levels showed similar changes after sevoflurane inhalation both with and without chemogenetic inhibition (*Figure 7D*, *Figure 7—source data 1*), suggesting that peripheral CRH levels changed with the same direction of PVH[CRH] neuronal activity.

Considering that corticosterone (CORT) is a hormone commonly used to measure stress in rodents (*Gong et al., 2015*), we measured the serum CORT levels and found that 30 min of 1.6% sevoflurane inhalation prominently evoked increased serum CORT concentrations after 1 hr termination of sevoflurane compared to that in groups inhaling pure oxygen (both groups injected saline or CNO), which were restored to normal levels (*Figure 7E*, *Figure 7—source data 1*). However, although the serum CRH levels of pure oxygen groups decreased after CNO injection, there was no significant difference in serum CORT concentrations between saline and CNO injection groups after inhalation of pure oxygen. These results implicate that the inhibition of PVH[CRH] neurons lowers the high levels of serum CORT due to sevoflurane exposure.

Next, we observed the stress-related behavior after inhaling pure oxygen (O$_2$) or 1.6% sevoflurane (sevo) for 30 min (*Figure 7B*). Analysis of self-grooming behavior (1.5 hr after sevoflurane GA

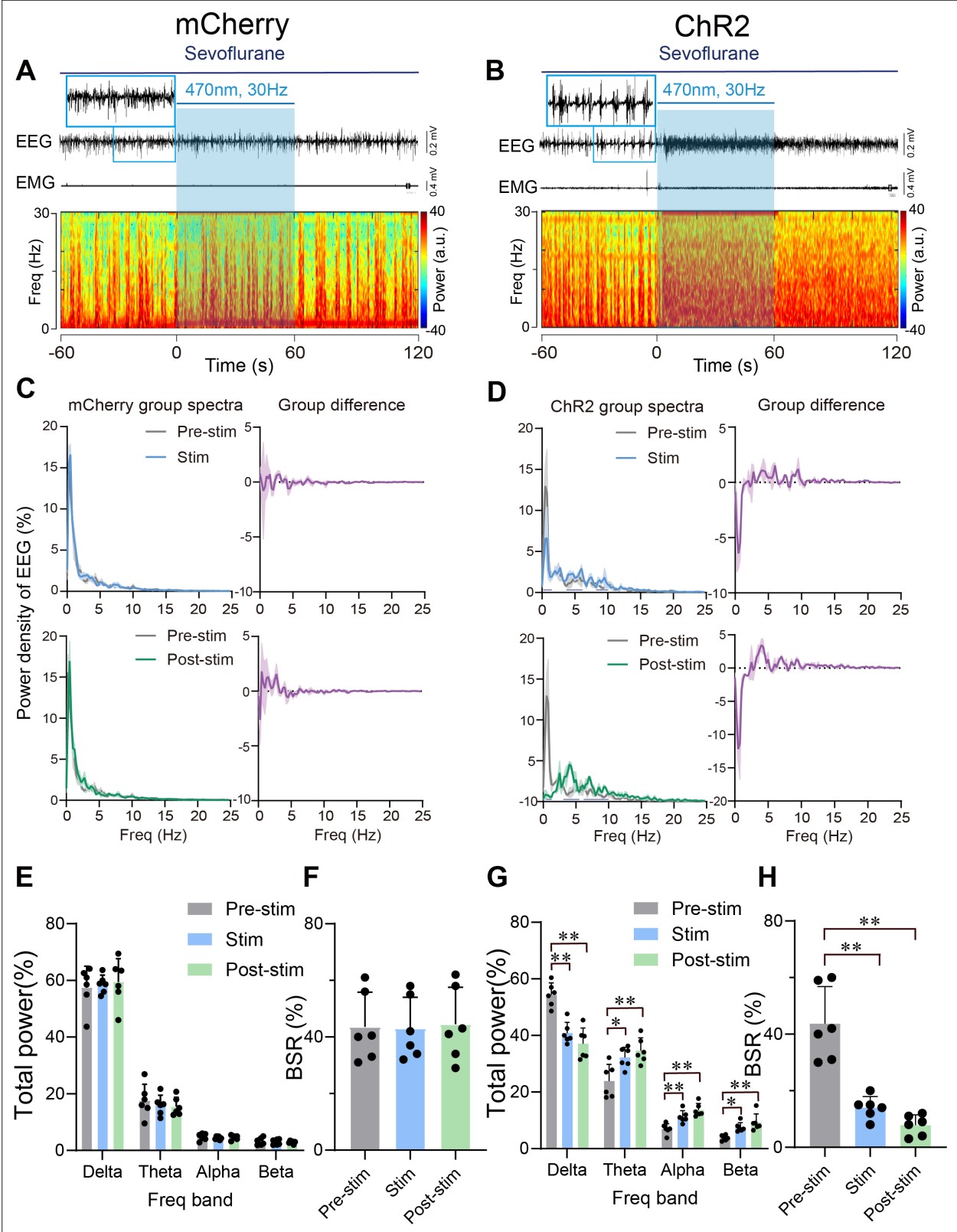

**Figure 6.** Optogenetic stimulation of PVH^CRH neurons induced cortical activation during burst-suppression oscillations induced by deep sevoflurane general anesthesia. Typical examples of electroencephalogram (EEG), electromyography (EMG), and EEG power spectral in a mouse injected with AAV-DIO-mCherry (**A**) or AAV-DIO-ChR2-mCherry (**B**) following acute photostimulation (30 Hz, 5 ms, 60 s) during burst-suppression oscillations. Time 0 indicates the beginning of photostimulation. The blue shadow indicates the 60 s duration of blue light stimulation. Top left: normalized group

*Figure 6 continued on next page*

*Figure 6 continued*

power spectral densities from PVH^CRH-mCherry mice (**C**) and PVH^CRH-ChR2 (**D**) with pre photostimulation (gray) and photostimulation (blue); top right: differences between normalized group power spectral densities from PVH^CRH-mCherry mice (**C**) and PVH^CRH-ChR2 mice (**D**). Bottom left: normalized group power spectral densities from PVH^CRH-mCherry mice (**C**) and PVH^CRH-ChR2 mice (**D**) with pre photostimulation (gray) and post photostimulation (green); bottom right: differences between normalized group power spectral densities from PVH^CRH-mCherry mice (**C**) and PVH^CRH-ChR2 mice (**D**). Dark blue lines in panel D indicate the power band with significant difference. Power percentage changes in cortical EEG before (gray), during (blue), and post (green) photostimulation in PVH^CRH-mCherry (**E**) or PVH^CRH-ChR2 (**G**) mice during burst-suppression oscillations. (**F, H**) BSR change before (gray), during (blue), and post (green) photostimulation in PVH^CRH-mCherry (**G**) or PVH^CRH-ChR2 (**H**) mice during burst-suppression oscillations ($n = 6$, two-way ANOVAs followed by Sidak's test, *$p < 0.05$, **$p < 0.01$; Pre-stim vs. Stim, $p < 0.01$, df = 15, 95% CI = 17.36–41.97; Pre-stim vs. Post-stim, $p < 0.01$, df = 15, 95% CI = 23.53–48.14). Stim, stimulation; Freq, frequency.

The online version of this article includes the following source data and figure supplement(s) for figure 6:

**Source data 1.** Original data for analysis displayed in *Figure 6E-H*.

**Figure supplement 1.** Optogenetic stimulation of PVH^CRH neurons induced cortical activation during burst-suppression oscillations induced by deep sevoflurane general anesthesia.

cessation) showed that the percentage of time spent self-grooming increased after 1.6% sevoflurane exposure (*Figure 7F*, *Figure 7—source data 1*). To validate the higher level of stress associated with sevoflurane GA, we conducted a series of tests to measure stress levels, including the open field test (OFT) (*Figure 7G*) and the elevated plus-maze test (EPM) (*Figure 7H*). Combining the results of the tests confirmed that the post-anesthesia period is associated with a high level of stress. To further determine the influence of chemogenetic inhibition of PVH^CRH neurons on the post-anesthesia stress response, we also performed stress-related behavioral testing on CRH-Cre mice expressing hM4Di receptors ($n = 8$ mice) in the PVH 30 min after i.p. administration of 3 mg/kg CNO or vehicle (*Figure 7A*). We observed that after 30 min of 1.6% sevoflurane exposure, mice moved lower percentage of distances in the central area during the OFT (*Figure 7G and I*, *Figure 7—source data 1*) and spent less time in the open arms during the EPM test (*Figure 7H and I*, *Figure 7—source data 1*) compared to the pure oxygen inhalation group. Intriguingly, PVH^CRH neuron-inhibited mice showed similar central distance and time spent in the open arms to those of the pure oxygen inhalation group, indicating that chemogenetic inhibition of PVH^CRH neurons abolished the stress response after sevoflurane GA. To further explore whether mice developed post-operative delirium induced by sevoflurane GA, which is characterized by disturbances in attention, awareness, and cognition (*Xu et al., 2021a*; *Peng et al., 2016*), we conducted the Y-maze test and novel object recognition test to test the cognitive function. We found that chemogenetic inhibition of PVH^CRH neurons ameliorated the short-term memory impairment caused by 30-min exposure to sevoflurane GA (*Figure 7—figure supplement 1*, *Figure 7—figure supplement 1—source data 1*), suggesting PVH^CRH neurons may involve in modulating sevoflurane-induced postoperative delirium.

## Discussion

Here, applying cutting-edge techniques, we found that PVH^CRH neuronal activity decreased during sevoflurane induction and gradually increased during the emergence phase, followed by hyperactivities with increased stress levels in mice after termination of sevoflurane administration. Furthermore, the modulation of PVH^CRH neurons bidirectionally altered the induction and recovery of sevoflurane GA. More importantly, chemogenetic inhibition of this population attenuated the post-anesthesia stress response, supporting the role of PVH^CRH neurons in mediating states of consciousness and stress response associated with sevoflurane GA.

The reversible loss of consciousness induced by sevoflurane GA has been reported to be the result of mutual regulation of numerous nuclei, including several wake-promoting ensembles (*Bao et al., 2023*), such as dopaminergic neurons in the ventral tegmental area (*Song et al., 2022*; *Gui et al., 2021*; *Taylor et al., 2016*; *Kenny et al., 2015*), glutamatergic neurons in the PVT (*Li et al., 2022*) and PB (*Wang et al., 2019*; *Xu et al., 2020a*), and neurons expressing dopamine D1 receptors (D1R) in the NAc (*Bao et al., 2021*), as well as sleep-promoting populations, such as GABAergic neurons in the rostromedial tegmental nucleus (*Vlasov et al., 2021*; *Yang et al., 2018*). In our study, we identified a new brain region involved in sevoflurane GA – PVH^CRH neurons – the activation of which prolongs the induction time and shortens the emergence time of sevoflurane GA, and modulates the neural inertia

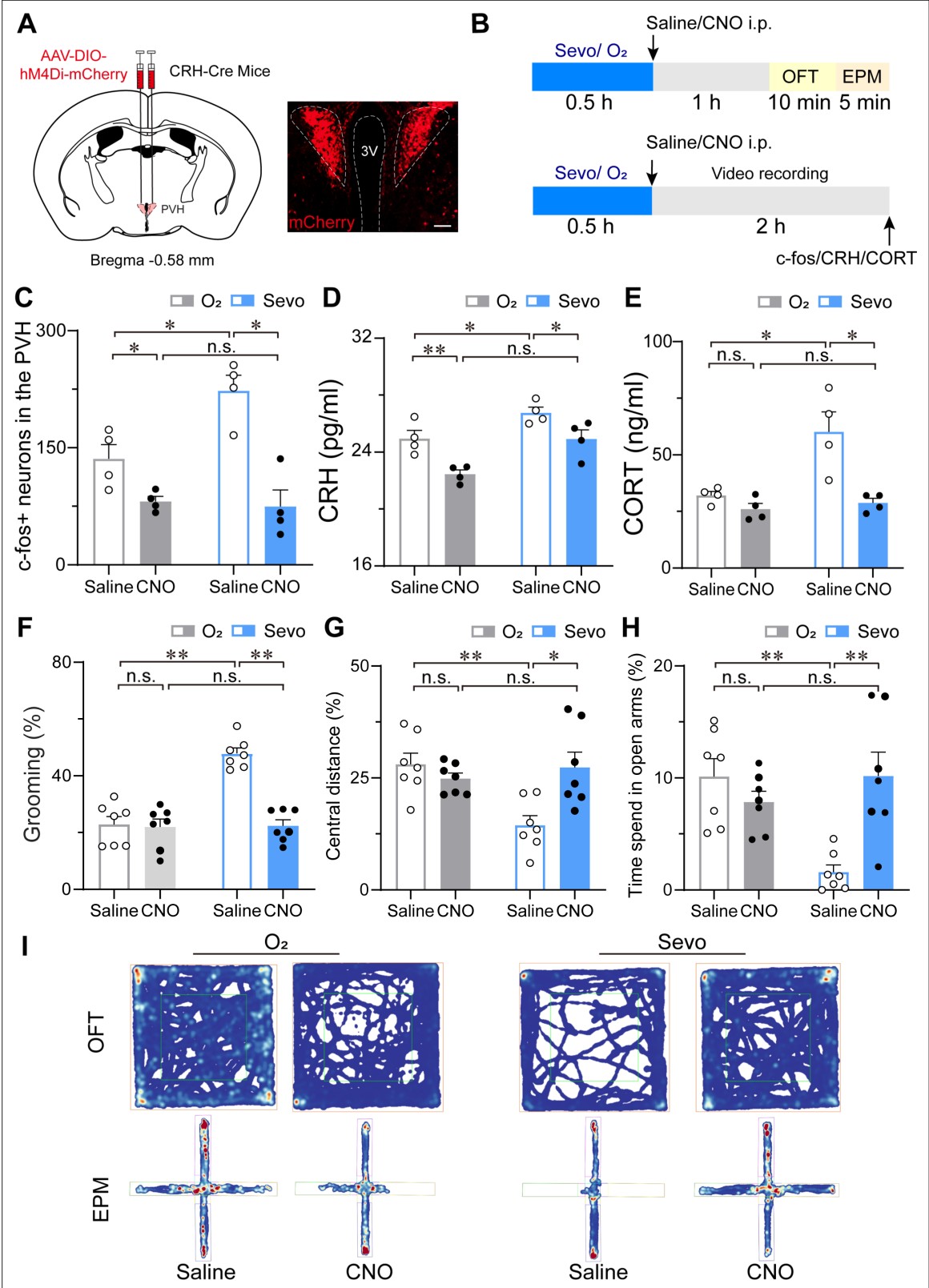

**Figure 7.** Chemogenetic inhibition of PVH^CRH neurons alleviated the stress response after sevoflurane GA. (**A**) Left: schematic of AAV-DIO-hM4Di-mCherry injected into the PVH of CRH-Cre mice. Right: representative image of mCherry immunofluorescence in the PVH. scale bar, 200 μm. (**B**) Experimental timeline of sacrificing mice for video recording, c-fos quantification, CRH, and CORT measurement or behavior testing (open field test [OFT] and elevated plus-maze [EPM]) after inhalation of sevoflurane or pure oxygen for 30 min. (**C-H**) Quantification of the number of c-fos-positive

*Figure 7 continued on next page*

*Figure 7 continued*

neurons in the PVH (**C**, $n = 4$, $F(1, 12) = 14.62$, saline: $O_2$ vs. sevo, $p = 0.0017$, $t = 3.523$, df = 12, 95% CI = −164.6 to −9.424; clozapine-*N*-oxide (CNO): $O_2$ vs. sevo, $p = 0.7122$, $t = 0.2632$, df = 12, 95% CI = −71.08 to 84.08), serum CRH (**D**, $n = 4$, $F(1, 12) = 0.4513$, saline: $O_2$ vs. sevo, $p = 0.0456$, $t = 2.604$, df = 12, 95% CI = −3.585 to −0.03; CNO: $O_2$ vs. sevo, $p = 0.0077$, $t = 3.567$, df = 12, 95% CI = −4.255 to −0.705) and CORT levels (**E**, $n = 4$, $F(1, 12) = 5.691$, saline: $O_2$ vs. sevo, $p = 0.0082$, $t = 3.608$, df = 12, 95% CI = −42.27 to −7.799; CNO: saline: $O_2$ vs. sevo, $p = 0.9144$, $t = 0.3853$, df = 12, 95% CI = −20.99 to 15.55), time percentage of self-grooming (**F**, $n = 7$, $F(1, 24) = 24.61$, saline: $O_2$ vs. sevo, $p < 0.001$, $t = 7.033$, df = 24, 95% CI = −34.57 to −16.98; CNO: $O_2$ vs. sevo, $p = 0.9275$, $t = 0.3486$, df = 24, 95% CI = −10.07 to 7.518), percentage of moving distances in the central areas of OFT (**G**, $n = 7$, $F(1, 24) = 6.953$, saline: $O_2$ vs. sevo, $p = 0.0018$, $t = 3.738$, df = 24, 95% CI = 4.906 to 21.86; CNO: $O_2$ vs. sevo, $p > 0.9999$, $t = 0.007$, df = 24, 95% CI = −8.799 to 8.747) and time percentage of staying in the open arms of the EPM (**H**, $n = 7$, $F(1, 24) = 12.70$, saline: $O_2$ vs. sevo, $p = 0.0017$, $t = 3.885$, df = 24, 95% CI = 3.198 to 13.62; CNO: $O_2$ vs. sevo, $p = 0.3783$, $t = 1.289$, df = 24, 95% CI = −8.972 to 2.717) following the protocol in (**B**). Statistical comparisons were conducted using two-way ANOVA followed by Sidak's tests. *p < 0.05, **p < 0.01, n.s., no significant differences. (**I**) Representative heatmaps of OFT and EPM after inhalation of sevoflurane or pure oxygen and administration of saline or CNO.

The online version of this article includes the following source data and figure supplement(s) for figure 7:

**Source data 1.** Original data for analysis displayed in *Figure 7C-H*.

**Figure supplement 1.** PVH$^{CRH}$ neurons involved in modulating sevoflurane-induced short-term memory impairment.

**Figure supplement 1—source data 1.** Original data for analysis displayed in *Figure 7—figure supplement 1*.

of the central nervous system (a tendency to resist behavioral state transitions between conscious and unconscious states) (*Friedman et al., 2010*). EEG is used to monitor the depth of anesthesia, and the EEG activity is linked to behavioral states of GA (*Shalbaf et al., 2015*). Burst-suppression is a phenomenon that occurs at deep anesthesia and is caused by the interaction of the hyperexcitable cortex with a refractory phase after the burst (*Ferron et al., 2009*; *Nguyen and Postnova, 2021*). This hyperexcitability is favored by reduced inhibition of anesthetics (*Ferron et al., 2009*). Notably, photo-stimulation of PVH$^{CRH}$ neurons effectively elicited cortical activation and behavioral emergence during steady sevoflurane GA state but only caused cortical activation during burst-suppression oscillations. Compared to the potent effect of optical stimulation of NAc D1R neurons during burst-suppression (*Bao et al., 2021*), our results indicate that optogenetic activation of PVH$^{CRH}$ neurons is insufficient to reverse the reduced inhibition. Furthermore, similar to the 'Arousal-Action Model' (*Liu and Dan, 2019*), GA-inhibiting neurons can also be classified into two categories referred to as 'arousal neurons' and 'action neurons.' Activation of the arousal neurons promotes EEG desynchronization but is insufficient to cause EMG activation during GA. In contrast, stimulation of action neurons causes both EEG desynchronization and behavioral emergence during GA. Based on this model, PVH$^{CRH}$ neurons can be classified as the arousal neurons, while NAc D1R neurons belong to the action neurons. Furthermore, it has been reported that preoperative anxiety increases the anesthetic requirements intraoperatively and prolongs recovery from anesthesia (*Kil et al., 2012*; *Wang et al., 2021b*), indicating that stress may induce abnormal neuronal activity, which influences the effect of anesthetic drugs and alters the induction of and emergence from GA. Given the vital role of PVH$^{CRH}$ neurons in the stress response regulation, it is also possible that PVH$^{CRH}$ neurons modulate sevoflurane GA through influencing stress levels.

The postoperative stress response is commonly associated with GA combined with surgery trauma. HPA activation during and after surgeries can result from multiple factors aside from anesthesia, including pain, anxiety, inflammation. Effective modulation of the stress response is of great importance in reducing the incidence of complications. As a predominant presentation of the post-anesthesia stress response, sevoflurane-induced EA is common, with an incidence of up to 80% in clinical reports, and is characterized by hyperactivity, confusion, delirium, and emotional agitation (*Dahmani et al., 2010*; *Lim et al., 2016*). Such responses pose the risk of neurotoxicity, delayed neurological development, and even cognitive deficits. However, a broad consensus on animal models mimicking this post-anesthesia stress response has yet to be established. Stress often leads to grooming and other repetitive behaviors, such as digging and circling in rodents (*Troisi, 2002*; *Langen et al., 2011*), and self-grooming is frequently observed and believed to contribute to post-stress de-arousal with adaptive value (*Song et al., 2016*; *Kalueff et al., 2016*). In this study, we observed increased self-grooming and a series of other stress-related behavioral parameters during the post-anesthesia period, combined with an elevated serum CORT level, which may indicate the high stress level and support these measures as indicators of post-anesthesia stress responses in animal models. Of note, the hyperactivity of PVH$^{CRH}$ neurons and behavior (e.g., excessive self-grooming) in mice could partly

recapitulate the observed agitation and underlying mechanisms during the emergence from sevoflurane GA in patients. Our findings may further help us to understand the underlying mechanism related to post-anesthesia stress responses observed in patients, leading to the development of strategies to prevent or treat the post-anesthesia stress response. However, considering the differences in species, duration of exposure, and surgical intensity could account for the observed discrepancies, our results may not directly translate to human surgical procedures.

Of particular note, the PVH is canonically viewed as the key endocrine controller of the stress response (*Denver, 2009*) and complex behavior after stress (*Bressers et al., 1995*; *Füzesi et al., 2016*), including self-grooming (*Kruk et al., 1998*). A recent study found that this role of the PVH was instructed by glutamatergic inputs from the lateral hypothalamus (LH) (*Mangieri et al., 2018*). Moreover, multiple lines of evidence have also demonstrated that PVH$^{CRH}$ neurons are not only responsible for launching the endocrine component of the mammalian stress response (*Denver, 2009*) but also for orchestrating post-stress behavior, including grooming (*Füzesi et al., 2016*). For example, empirical evidence proposed that the 'PVH$^{CRH}$-the caudal part of the spinal trigeminal nucleus-spinal' descending pathway translates stress-related stimuli into motor actions of self-grooming (*Xie et al., 2022*). In our study, we observed elevated PVH$^{CRH}$ neuronal activity along with excessive self-grooming during the post-anesthesia period. Inspiringly, chemogenetic inhibition of PVH$^{CRH}$ neurons significantly alleviated these responses, underscoring the crucial role of PVH$^{CRH}$ neurons in sevoflurane GA-induced self-grooming. Across the whole GA process, PVH$^{CRH}$ neurons were greatly suppressed under sevoflurane GA but were recruited to control stress responses with rebound excitation after cessation of sevoflurane GA, and therefore evoked self-grooming and other behaviors to facilitate the reduction in stress level. Therefore, our data contribute to an emerging view that PVH$^{CRH}$ neurons also play a key role in mediating the post-anesthesia stress responses.

Interestingly, we found that sevoflurane GA-induced self-grooming exhibited exclusive features different from other grooming models, and the LH, which provides glutamatergic inputs to the PVH in grooming (*Mangieri et al., 2018*), was not activated during the post-anesthesia period (*Figure 1—figure supplement 2B*), implying the existence of other functional pathways. PVH$^{CRH}$ neurons received direct long-range GABAergic inputs from multiple brain regions, such as the LS, raphe magnus nucleus, and bed nucleus of the stria terminalis (*Yuan et al., 2019*). Meanwhile, it has been demonstrated that the neuronal activity of the suprachiasmatic nucleus negatively regulates the activity of PVH$^{CRH}$ neurons by GABA release (*Ono et al., 2020*). This evidence provides a possible explanation for the hyperactivity of PVH$^{CRH}$ neurons in the post-anesthesia period, whereby sevoflurane GA suppresses activity of these brain areas, and therefore disinhibits PVH$^{CRH}$ neurons by decreasing GABAergic inputs. Further research is required to fully understand this mechanism.

In conclusion, we have identified a significant node modulating the effect of sevoflurane GA in both during- and post-anesthesia stages: the PVH$^{CRH}$ neurons. These neurons play a potent role in modulating anesthesia states in sevoflurane GA and are part of sevoflurane's anesthesia regulatory network.

# Materials and methods

## Key resources table

| Reagent type (species) or resource | Designation | Source or reference | Identifiers | Additional information |
|---|---|---|---|---|
| Biological sample (*Mus musculus*) | CRH-Cre mice | South China Agricultural University/ the Shanghai Model Organisms Center | Crh < Cre > B6(Cg)-Crhtm1(cre)Zjh/J | JAX #012704 |
| Biological sample (*Mus musculus*) | C57BL/6 mice | The Experimental Animal Management Department, Institute of Family Planning Science, Shanghai | C57BL/6J | JAX #000664 |
| Biological sample (*Mus musculus*) | CD-1 mice | The Experimental Animal Management Department, Institute of Family Planning Science, Shanghai | Crl:CD1(ICR) | |
| Antibody | Rabbit polyclonal anti-c-fos antibody | Abcam, USA | ab190289 | IF (1:8000) |
| Antibody | Donkey polyclonal anti-rabbit Alexa488 secondary antibody | Abcam, USA | ab150073 | IF (1:1000) |

*Continued on next page*

*Continued*

| Reagent type (species) or resource | Designation | Source or reference | Identifiers | Additional information |
|---|---|---|---|---|
| Antibody | Rabbit monoclonal anti-CRF antibody | Abcam, USA | ab272391 | IF (1:800) |
| Recombinant DNA reagent | AAV-EF1α-DIO-hM3D(Gq)-mCherry-WPRE | Brain VTA, China | Cat#.PT-0988 | $3.3 \times 10^{12}$ vector genomes (VG)/ml |
| Recombinant DNA reagent | AAV-EF1α-DIO-mCherry-WPRE-hGH pA | Brain VTA | Cat#.PT-0013 | $5.35 \times 10^{12}$ VG/ml |
| Recombinant DNA reagent | AAV2/9-hEF1α-DIO-hM4D(Gi)-mCherry-pA | Taitool Bioscience, China | S1060−1x2 | $3.3 \times 10^{12}$ VG/ml |
| Recombinant DNA reagent | AAV2/9-hSyn-DIO-jGCaMP7b-WPRE-pA | Taitool Bioscience, China | S0607-9 | $1.61 \times 10^{13}$ VG/ml |
| Recombinant DNA reagent | AAV2/9-hEF1a-DIO-hChR2-mCherry-pA | Taitool Bioscience, China | S0199-9 | $1.61 \times 10^{13}$ VG/ml |
| Recombinant DNA reagent | AAV2/9-hEF1a-DIO-mCherry-pA | Taitool Bioscience, China | S0506-9 | $1.61 \times 10^{13}$ VG/ml |
| Recombinant DNA reagent | AAV-CAG-FLEX-taCasp3-TEVp | Brain VTA | PT-0206 | $5.88 \times 10^{12}$ VG/ml |
| Commercial assay or kit | Enzyme-linked immunosorbent assay (ELISA) kits | CUSABIO Technology, China | CSB-E14068m and CSB-E07969m | |
| Chemical compound, drug | Pentobarbital sodium | Protein Biotechnology Co., Ltd | CAS-57-33-0 | 45 mg/kg |
| Chemical compound, drug | Sevoflurane | Lunan BETTER Pharmaceutical Co., Ltd | H20233956 | 1.2%, 1.6%, or 2.0% |
| Chemical compound, drug | Clozapine-N-oxide | LKT, USA | C4759 | 3 mg/kg |
| Chemical compound, drug | Colchicine | MedChemExpress | CAS-64-86-8 | 20 µg in 500 nl saline |
| Software, algorithm | Spike2 | Cambridge, UK | Spike2 1.0 | CED |
| Software, algorithm | SleepSign | Kissei omtec, Japan | SleepSign 3.0 | |
| Software, algorithm | Power meter | Coherent, USA | PM 10 | |
| Software, algorithm | Automatic video tracking system | Shanghai Vanbi Intelligent Technology Co., Ltd | Tracking Master V3.0 | |
| Software, algorithm | GraphPad Prism | GraphPad Software, USA | GraphPad Prism 8.0 | |
| Sequence-based reagent | *Crh*-common | Beijing Tsingke Biotech Co., Ltd | PCR primers | CTT ACA CAT TTC GTC CTA GCC |
| Sequence-based reagent | *Crh*-wt | Beijing Tsingke Biotech Co., Ltd | PCR primers | CAC GAC CAG GCT GCG GCT AAC |
| sequence-based reagent | *Crh*-mutant | Beijing Tsingke Biotech Co., Ltd | PCR primers | CAA TGT ATC TTA TCA TGT CTG GAT CC |

## Mice

CRH-Cre mice (Crh < Cre > B6(Cg)-Crh^tm1(cre)Zjh/J, JAX #012704) were kindly provided by Prof. Gang Shu from South China Agricultural University and were obtained from the Shanghai Model Organisms Center. Wild-type C57BL/6 and CD-1 mice were obtained from the Experimental Animal Management Department, Institute of Family Planning Science, Shanghai. Mice were group-housed in a soundproof room, with an ambient temperature of 22 ± 0.5°C and a relative humidity of 55% ± 5%. The mice were provided with adequate amounts of food and water, and were housed under an automatic 12 hr light/12 hr dark cycle (illumination intensity 100 lux, lights on at 07:00). Adult C57BL/6 male mice (CRH-Cre and wild-type mice, 8–10 weeks old, 20–30 g) were used for all experiments; adult CD-1 male mice (4–6 months old, 40–50 g) were used for the direct attack-induced model. All behavioral experiments were performed during the daytime (7:00–19:00). Animals were randomly divided into the experimental groups and control group. CRH-Cre mice were used in calcium fiber photometry recordings, chemogenetic and optogenetic manipulations, and wild-type mice were

used for the identification of active neurons and the characterization of self-grooming during the post-anesthesia period. All experimental procedures were approved by the Animal Care and Use Committee of Fudan University (certificate number: 20160225-073). All efforts were made to minimize animal suffering and discomfort and to reduce the number of animals required to produce reliable scientific data.

## Surgery

Mice were anesthetized with 2% pentobarbital sodium (45 mg/kg) by intraperitoneal injection and placed on a stereotaxic frame (RWD Life Science, China). The temperature of each mouse was maintained constant using a heading pad during the operation. After shaving the head and sterilizing with 75% ethanol, the parietal skin was cut along the sagittal suture and small craniotomy holes were made above the PVH. For chemogenetic experiment, the virus containing recombinant AAV-EF1α-DIO-hM3D(Gq)-mCherry-WPRE ($3.3 \times 10^{12}$ vector genomes (VG)/ml, Brain VTA, China), or AAV2/9-hEF1α-DIO-hM4D(Gi)-mCherry-pA ($3.3 \times 10^{12}$ VG/ml, Taitool Bioscience, China), or a viral vector encoding Cre-dependent expression of Caspase-3 (AAV-CAG-FLEX-taCasp3-TEVp, $5.88 \times 10^{12}$ VG/ml) was bilaterally injected into the PVH (anteroposterior [AP] = –0.4 to 0.5 mm, mediolateral [ML] = ±0.2 mm, dorsoventral [DV] = 4.5 mm deep relative to endocranium) via a glass injection needle and an air-pressure-injector system at a rate of 0.1 μl/min. Control mice received injections of the control virus AAV-EF1α-DIO-mCherry-WPRE-hGH pA ($5.35 \times 10^{12}$ VG/ml, Brain VTA, China) at the same coordinates and volumes. After 7 min of injection to each site, the glass pipette was left at the injection site for 8 min before slowly withdrawing.

For in vivo fiber photometry recordings and optogenetic experiments, after injecting the virus (AAV2/9-hSyn-DIO-jGCaMP7b-WPRE-pA, AAV2/9-hEF1a-DIO-hChR2-mCherry-pA, or AAV2/9-hEF1a-DIO-mCherry-pA, $1.61 \times 10^{13}$ VG/ml, Taitool Bioscience, China) into the bilateral PVH at a rate of 0.1 μl/min for 7 min, the optical fiber cannula (outer diameter [OD] = 125 mm, inner diameter = 400 μm, numerical aperture [NA] = 0.37; Newdoon, Shanghai, China) was placed above the unilateral PVH ([AP] = –0.4 to 0.5 mm, [ML] = +0.2 mm, [DV] = 4.0 mm deep relative to the endocranium) before implanting the EEG and EMG electrodes. The cannulas and the EEG and EMG electrodes were secured to the skull with dental cement. Mice were then kept in a warm environment until they resumed normal activity and fed carefully for 2 weeks before the following experiments.

## Calcium fiber photometry recordings

Using a multichannel fiber photometry system equipped with 488 nm laser (OBIS 488LS; Coherent) and a dichroic mirror (MD498; Thorlabs), the fluorescence signals of the GCaMP were collected by a photomultiplier tube (R3896, Hamamatsu) and recorded using a Power1401 digitizer and Spike2 software (CED, Cambridge, UK) with simultaneous EEG/EMG recordings. During this experiment, mice were first placed in a cylindrical acrylic glass chamber connected to a sevoflurane vaporizer (VP 300; Beijing Aeonmed, China). The concentration of sevoflurane was monitored by an anesthesia monitor (Vamos, Drager, China) connected to the chamber. In the dose-dependent experiments, after recording the baseline signals of awake mice during a 20-min adaption period, different concentrations of sevoflurane (1.2%, 1.6%, or 2.0%) were continuously delivered by 1.5 l/min flow of 100% oxygen for 20 min. The group inhaling 100% oxygen was set as the control group. As for the observation of the $Ca^{2+}$ signals during the post-anesthesia period, the baseline signals of awake mice were recorded for 30 min before 30 min delivery of 1.6% of sevoflurane. After turning off the sevoflurane vaporizer, the recording lasted until the $Ca^{2+}$ signals returned to the baseline level.

## Chemogenetic manipulations

All the chemogenetic experiments were performed between 19:00 and 24:00, with at least a 24-hr washout time. After intraperitoneally pretreating the animals with saline or 3 mg/kg of CNO (C4759, LKT, USA), the mice were placed in a cylindrical acrylic glass chamber with 1.5 l/min flow of 100% oxygen for 20 min for habituation. To determine the sensitivity of sevoflurane, mice experienced stepwise increasing exposure to sevoflurane, initially delivered at 1.2% and increased every 10 min in increments of 0.1%. The righting reflex of each animal was assessed after 10 min at each concentration, ensuring both brain and chamber balance. If a mouse successfully returned to the prone position twice in a row, then it was considered to have a complete righting reflex. Otherwise, the righting

reflex was determined not to be present. The induction and emergence experiments were conducted as follows. Each mouse was intraperitoneally administered saline or CNO (3 mg/kg, i.p.) 1 hr before the experiment. Next, 30 min 2% sevoflurane was delivered with 1.5 l/min 100% oxygen after 20-min habituation. Following exposure to 2% sevoflurane for 30 min, the mouse was removed from the glass chamber and kept in the supine position in the indoor air. The time to induction was defined as the interval between the time when the mouse was put into the glass chamber full of 2% sevoflurane to the time when the mice exhibited LORR. The time to emergence was defined as the interval between the time when the mouse was removed from the chamber to the time when it regained the righting reflex with all four paws touching the floor. Throughout the entire experimental period, an electric blanket was put under the chamber to maintain the chamber temperature at 36°C.

## Optogenetic stimulation

Optical stimulation was first performed during induction of and recovery from sevoflurane GA. After recording an awake EEG/EMG baseline for 10 min (continuously inhaling pure oxygen), we increased the concentration of sevoflurane in the chamber to 2.0% and applied light-pulse trains (5 ms pulses at 30 Hz) simultaneously until LORR. Next, after each mouse inhaled 2.0% sevoflurane for 30 min, we turned off the sevoflurane vapor and initiated acute light stimulation (5 ms pulses at 30 Hz) until RORR.

For manipulation during the steady sevoflurane GA state, light-pulse trains (473 nm, 5 ms pulses at 30 Hz for 60 s) were applied according to a previous protocol (*Taylor et al., 2016*). In brief, a baseline EEG/EMG was recorded after habituation in the acrylic glass chamber for 10 min. Next, the sevoflurane vapor was turned on and the mice were subjected to sevoflurane with an initial 2.5% sevoflurane for 20 min and then placed in a supine position and then the sevoflurane concentration was reduced to 1.4%. If the animals showed any signs of LORR, the concentration was increased by 0.1% until the LORR maintained at a constant concentration for at least 20 min. Photostimulation (5 ms pulses at 30 Hz for 60 s) was applied through a laser stimulator (SEN-7103, Nihon Kohden, Japan) when EEG patterns showed characteristics of steady sevoflurane GA state (low-frequency, high-amplitude activity) for at least 5 min. As for the burst-suppression period, a baseline EEG/EMG was recorded during habituation in the acrylic glass chamber for 10 min. Next, we delivered 2% sevoflurane with pure oxygen at a speed of 1.5 l/min for induction initially. After entering a continuous and steady burst-suppression oscillation mode for at least 5 min, we applied blue-light stimulation (5 ms pulses at 30 Hz) for 60 s. A power meter (PM10, Coherent, USA) was used to measure the power intensity of the blue light before experiments.

## EEG/EMG analysis

SleepSign software (Kissei Comtec, Japan) was used to record and analyze all EEG/EMG signals. EEG/EMG signals were magnified and filtered (EEG, 0.5–30 Hz; EMG, 20–200 Hz), and were digitized at a 128 Hz sampling rate. Subsequently, fast-Fourier transformation was used to calculate the EEG power spectra for successive 4 s epochs in the frequency range of 0–25 Hz. EEG frequency bands (delta, 0.5–4.0 Hz; theta, 4.0–10 Hz; alpha, 10–15 Hz; beta, 15–25 Hz) were established in view of previous research, and relative variations in aggregate power were calculated. In the optogenetic experiments during steady sevoflurane GA state, a total of 120 s EEG signals were analyzed (before and during acute light stimulation). As for optogenetic tests during burst-suppression oscillations, a total of EEG signals from 180 s epochs were calculated (before, during, and after acute light stimulation). The BSR was calculated by the percentage of suppression in the time spans of 1 min before, during, or after photostimulation. Raw EEG data recorded by SleepSign software were converted to text format for further analysis of BSR using MATLAB as demonstrated previously (*Bao et al., 2021*). The minimum duration of burst and suppression periods was set to 0.5 s.

## Immunohistochemistry

The immunohistochemistry test was conducted according to the previous studies (*Luo et al., 2018a*). To confirm the correct injection sites, after finishing all related experiments, each mouse was anesthetized with chloral hydrate (360 mg/kg) and then perfused intracardially with 30 ml phosphate-buffered saline (PBS) followed by 30 ml 4% paraformaldehyde (PFA). The brains of the mice were removed and postfixed in 4% PFA overnight and then incubated in 30% sucrose phosphate buffer at 4°C until they sank. Coronal sections (30 μm) were cut on a freezing microtome (CM1950, Leica, Germany), and

the fluorescence of injection sites was examined to locate the PVH according to the histology atlas of Paxinos and Franklin (2001, The Mouse Brain in Stereotaxic Coordinates 2nd edn [San Diego, CA: Academic]) using a microscope (Fluoview 1200, Olympus, Japan).

For immunostaining of c-fos, mice were sacrificed at set time according to different experiments and perfused with PBS followed by 4% PFA in 0.1 M phosphate buffer. The brain was then dissected and fixed in 4% PFA at 4°C overnight. Fixed samples were sectioned into 30 μm coronal sections using a freezing microtome (CM1950, Leica, Germany). Brain sections were washed in PBS three times (5 min each time) and incubated for 48 hr with primary antibody in PBST (containing 0.3% Triton X-100) on a 4°C agitator using rabbit anti-c-fos (1:8000, Abcam, USA) antibody, or rabbit anti-CRF (1:800, Immunostar 20084, bilateral intracerebroventricular infusion of colchicine (total 20 μg in 500 nl saline, coordinates from Bregma: AP = –0.22 mm, ML = ±1.15 mm, DV = 2.06 mm) 24 hr before perfusion) (*Antoni et al., 1983*). Next, brain sections were washed in PBS three times (5 min each time) and incubated with a donkey-anti-rabbit Alexa488 secondary antibody (1:1000, Abcam, USA) in PBST for 2 hr at room temperature. Finally, the sections were mounted on glass slides, dried, dehydrated, and cover-slipped. Fluorescence images were collected with an Olympus VS120 automated slide scanner. Only data from mice in whom the AAV infection and optical fiber location were confirmed were included. The number of animals that were excluded because of poor virus expression or misplaced optical fibers are as follow: jGCaMP7b, $n = 5$; hM3Dq, $n = 7$; hM4Di, $n = 10$; mCherry, $n = 5$; ChR2, $n = 12$; wrong optical fiber location, $n = 12$.

## Behavior experiments
### Righting reflex assessment
GA was defined by the loss of the righting reflex, a widely used surrogate measure in rodents. All mice were placed in the air-tight, temperature-controlled, 200 ml cylindrical chambers with 100% oxygen flowing at a speed of 1.5 l/min for 10 min for the sake of habitation. To determine the $EC_{50}$ of LORR, the initial concentration of sevoflurane was set at 1.0%, and the concentrations were increased by 0.1% every 15 min. During the last 2 min of every 15 min, the chambers were rotated 180° to observe whether the righting reflex of the mice was present, and the number of animals that lost their righting reflex at each concentration was recorded. To determine the $EC_{50}$ of RORR, all the mice were anesthetized with 2.0% sevoflurane for 30 min, and the concentration was decreased in decrements of 0.1% until all the mice had recovered their righting reflex. The number of mice that had recovered their righting reflex at each concentration was counted. Mice experienced stepwise increasing and diminishing exposure to sevoflurane, with 15 min taken at each step, during which and an electric blanket was used to maintain the chamber temperature at 36°C. The righting reflex of each animal was assessed after 15 min at each concentration, ensuring both brain and chamber balance. If a mouse successfully returned to the prone position twice in a row, then it was considered to have a complete righting reflex. Otherwise, the righting reflex was determined not to be present.

## Arousal scoring
Arousal responses were scored according to the methods of previous studies (*Bao et al., 2021*). Spontaneous movements of the limbs, head, and tail were scored by the movements' intensity as three levels: absent (0), mild (1), or moderate (2). Righting was scored as 0 if the mouse remained LORR, and 2 if the mouse recovered its righting reflex. Walking was scored as follows: 0 = no further movements; 1 = crawled without raising the abdomen off the chamber bottom; and 2 = walked with all four paws and the abdomen off the chamber bottom. The total score for each mouse was defined as the sum of all categories during the photostimulation period.

## Assessment of self-grooming
The behavior of the animals was recorded in a test chamber (cylindrical acrylic glass, 15 cm diameter, 20 cm height) in a quiet and sound-proof room by a video camera (HIKVISION, US). Mice were habituated in the test chamber for 20 min before testing. For the spontaneous grooming model, the mouse was placed into the test chamber and spontaneous activities were recorded over a 20-min period. For restraint stress-induced grooming, the mouse was restrained in a stainless-steel tube (5 cm diameter, 10 cm length) for 20 min and then placed immediately into the test chamber for 20 min video recording. For direct attack-induced grooming, a C57BL/6 mouse was placed directly into the home

cage of a larger and aggressive CD-1 mouse for 5 min with the aggressor present (*Golden et al., 2011*). Immediately afterward, the C57BL/6 mouse was placed in the test chamber, and its behavior was recorded for 20 min. For water spray-induced grooming (*Mangieri et al., 2018*), the mouse was sprayed with a spray bottle prefilled with sterile water (25°C) directed to the face, belly, and back, before placing the mouse in the test chamber for 20 min of video recording. For swimming-induced grooming, the mouse was placed into a plastic cylinder filled with water at 22–26°C and allowed them to freely swim for 5 min. Excess water was then removed before the animal was placed into the test chamber for video recording for 20 min.

The self-grooming behavior was manually quantified by observers blinded to the experimental conditions. Self-grooming was defined as when the mouse licked its own body parts, including the paws legs, tail, and genitals, or stroked its own nose, eyes, head, body, with the front paws (*Kalueff et al., 2016*). An interruption of 6 s or more separated two individual bouts (*Kalueff et al., 2007*; *Kalueff and Tuohimaa, 2005*). The number of grooming bouts, the mean bout duration, and the total grooming time spent in the test period were evaluated. For the analysis of the patterns and micro-structures of grooming behavior, the time spent on grooming different body parts and the transitions between different phases were calculated based on previous protocols (*Kalueff et al., 2007*; *Kalueff and Tuohimaa, 2005*; *Mu et al., 2020*). The following scaling systems based on grooming stages were used: no grooming (0), paw licking (1), nose/face/head wash (2), body grooming (3), leg licking (4), and tail/genitals grooming (5). Correct transitions were defined as the following progressive transitions between the grooming stages: 0–1, 1–2, 2–3, 3–4, 4–5, and 5–0. Incorrect transitions were characterized by skipped (e.g., 0–5, 1–5) or reversed (e.g., 3–2, 4–1, 5–2) stages. Interrupted grooming bout was considered if at least one interruption was recorded during the transition.

### Open field test

The OFT is one of the most widely used behavioral tests to evaluate anxiety-like behavior in rodents (*Mu et al., 2020*). In the OFT, a camera was used to record the movement of mice in a 50 × 50 cm area. The middle square (30 × 30 cm) of the box was considered the central area. The entire device was surrounded by 40 cm high walls and its inner surface was painted black. In a quiet and sound-proof room with dim light (≈20 lx), a mouse was placed in the middle of the arena and allowed to move freely for 10 min. An automatic video tracking system (Tracking Master V3.0, Shanghai Vanbi Intelligent Technology Co., Ltd) was used to record the ambulatory activity of the mice (*Zhong et al., 2022*). After each test, the apparatus was wiped clean with 75% alcohol to avoid the influence of the previous mouse. The total and central travel distances were quantified.

### Elevated plus-maze test

The EPM test is commonly used to measure anxiety-like behavior in mice (*Mu et al., 2020*). The device consists of two open arms (20 × 6 × 0.5 cm) and two closed arms (20 × 6 × 30 cm), which lie across each other with an intersection part of 6 × 6 × 0.5 cm. The open arms have insignificant walls (0.5 cm) to reduce the frequency of falls, whereas the closed arms are surrounded by high walls (30 cm). The whole device was 70 cm above the floor. The camera was placed above the EPM in the behavioral test room. Each mouse was placed in the behavioral testing room 1 hr before the test for adaption. During the test, the mouse was placed in the intersection area of the maze, facing an open arm. After exploring the maze freely for 5 min, the mouse was removed from the EPM and returned to its home cage. The maze was cleaned with a spray bottle and paper towel before testing the next mouse. An automatic video tracking system (Tracking Master V3.0, Shanghai Vanbi Intelligent Technology Co., Ltd) was used to analyze the movement data of the mouse. An entry was defined when half of the mouse's body entered an area.

### Y-maze test

The Y-maze test for evaluating learned behavior was conducted as previously described (*Xu et al., 2021a*; *Peng et al., 2016*). Initially, the test mouse was placed at the start of one arm, facing away from the center, and allowed to explore the apparatus freely for 5 min while another arm was blocked. After this period, the mouse was returned to its home cage. Subsequently, the block was removed, and the mouse was allowed to explore all arms freely for another 5 min. The discrimination index,

which is the time spent in the novel arm divided by the time spent in both arms, was used to assess spatial memory.

### Novel object recognition test

The novel object recognition test was conducted in the same arena as the OFT. During the habituation session, the test mouse was placed in the arena to explore for 10 min. Following habituation, two objects of similar size but different shape and color were placed in opposite corners of the arena. The test mouse was then placed in the center and allowed to explore for 10 min. After this exploration, all mice were exposed to 2.0% sevoflurane for 30 min and then returned to their home cages. Thirty minutes later, saline or CNO was injected. One hour after the injection, one of the objects was replaced, and the same test mouse was allowed to explore for another 10 min. Interactions with each object (defined as sniffing and/or having the head within 2–3 cm of the object) were recorded for analysis. Mice that did not reach a minimum of 20 s of exploration time for both objects during either of the two 10-min trials were excluded from analysis, as it could not be confirmed that they spent enough time exploring to learn or discriminate between the objects. The time spent interacting with the novel object, divided by the time spent interacting with both objects, was used to describe the preference for the novel object.

### Measurement of CRH and CORT levels

Mice were decapitated immediately at 1 hr after the mice received either saline or CNO (3 mg/kg, C2041, LKT) treatment, respectively. Blood was rapidly collected from the retroorbital plexus of the mice subjected to sevoflurane GA or pure oxygen. Considering the circadian rhythm release pattern of CORT, which has lower concentrations in the morning (*Kim et al., 2008*), blood sampling was conducted between 8:00 and 12:00 to minimize the effect of time of day on glucocorticoid levels. Blood samples were allowed to clot for 2 hr at room temperature before centrifugation for 15 min at $1000 \times g$. Serum was removed and assayed immediately or aliquoted, and samples were stored at −80°C. The concentrations of CRH and CORT in the serum were detected using enzyme-linked immunosorbent assay (ELISA) kits (CSB-E14068m and CSB-E07969m, CUSABIO Technology, China) according to the manufacturer's instruction.

### Quantification and statistical analysis

All data are presented as the mean ± standard deviation. All experimental data were analyzed in a blinded manner. The selection of sample sizes was based on similar experiments that use chemogenetic and optogenetic methods (*Wang et al., 2019*; *Bao et al., 2021*). Paired or unpaired two-tailed Student's *t*-test was selected to compare the two groups. If the data did not satisfy the normal distribution, further analysis was performed using Wilcoxon signed-rank tests or Mann–Whitney rank-sum tests. Multiple comparisons were made using one-way repeated-measures ANOVA with Tukey's post hoc test or two-way repeated-measures ANOVA followed by Sidak's post hoc comparison test. Statistical analysis was performed using GraphPad Prism 8.0 (GraphPad Software, USA). Full details of the statistical analyses are shown in the figure legends. $p < 0.05$ was statistically significant in all cases.

## Acknowledgements

We thank Prof. Gang Shu from South China Agricultural University for kindly providing CRH-Cre mice, and Yuan Meng, Mingxuan Yang, Ziheng Zhao, Yingfan Guo, and Yi Zhang for assisting experiments and data analysis. Funding Statement: Chang-Rui Chen, Wei-Min Qu, and Zhi-Li Huang supervised the study and acquired funding. This study was supported in part by: National Major Project of China Science and Technology Innovation STI2030-Major Project (2021ZD0203400 to Z-LH), National Key Research and Development Program of China (2021YFC2501400 to W-MQ), National Natural Science Foundation of China (82020108014 and 32070984 to Z-LH; 82071491 to W-MQ; 81671317 to C-RC), Shanghai Science and Technology Innovation Action Plan Laboratory Animal Research Project (201409001800), Program for Shanghai Outstanding Academic Leaders Shanghai Municipal Science and Technology Major Project and ZJLab (2018SHZDZX01).

# Additional information

## Funding

| Funder | Grant reference number | Author |
|---|---|---|
| National Science Foundation of China | STI2030-Major Projects-2021ZD0203400 | Zhi-Li Huang |
| National Key Laboratory Foundation of China | 2021YFC2501400 | Wei-Min Qu |
| National Natural Science Foundation of China | 82020108014 | Zhi-Li Huang |
| National Natural Science Foundation of China | 32070984 | Zhi-Li Huang |
| National Natural Science Foundation of China | 82071491 | Wei-Min Qu |
| National Natural Science Foundation of China | 81671317 | Chang-Rui Chen |
| Shanghai Science and Technology Development Foundation | 201409001800 | Zhi-Li Huang |
| Shanghai Municipal Youth Science and Technology Star Project | 2018SHZDZX01 | Zhi-Li Huang |

The funders had no role in study design, data collection, and interpretation, or the decision to submit the work for publication.

## Author contributions

Shan Jiang, Conceptualization, Data curation, Software, Formal analysis, Visualization, Writing – original draft, Writing – review and editing; Lu Chen, Data curation, Formal analysis, Validation, Visualization, Writing – review and editing; Wei-Min Qu, Funding acquisition; Zhi-Li Huang, Supervision, Funding acquisition, Writing – review and editing; Chang-Rui Chen, Conceptualization, Resources, Supervision, Funding acquisition, Writing – review and editing

## Author ORCIDs

Lu Chen ⬥ https://orcid.org/0009-0005-2106-4980
Zhi-Li Huang ⬥ https://orcid.org/0000-0001-9359-1150
Chang-Rui Chen ⬥ https://orcid.org/0000-0001-9879-9620

## Ethics

All experimental procedures were approved by the Animal Care and Use Committee of Fudan University (certificate number: 20200306-023). All efforts were made to minimize animal suffering and discomfort and to reduce the number of animals required to produce reliable scientific data.

Joint public review: https://doi.org/10.7554/eLife.90191.4.sa1
Author response https://doi.org/10.7554/eLife.90191.4.sa2

# Additional files

## Supplementary files
• MDAR checklist

## Data availability

All data generated or analyzed during this study are included in the manuscript and source data files.

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
