## [Editor Report · eLife assessment]

This study presents **useful** findings for how sevoflurane anesthesia modulates the activity of corticotropin-releasing hormone neurons in the paraventricular nucleus of the hypothalamus and how manipulation of such PVHCRH neurons influences anesthesia and post-anesthesia responses. The technical approaches are **solid** and the data presented is largely clear. Whether PVHCRH neurons are critical for the mechanisms of sevoflurane anesthesia is a direction for the future.

---

## [Referee Report · Joint public review]

This study describes a group of CRH-releasing neurons, located in the paraventricular nucleus of the hypothalamus, which, in mice, affects both the state of sevoflurane anesthesia and a grooming behavior observed after it. PVHCRH neurons showed elevated calcium activity during the post-anesthesia period. Optogenetic activation of these PVHCRH neurons during sevoflurane anesthesia shifts the EEG from burst-suppression to a seemingly activated state (an apparent arousal effect), although without a behavioral correlate. Chemogenetic activation of the PVHCRH neurons delays sevoflurane-induced loss of righting reflex (another apparent arousal effect). On the other hand, chemogenetic inhibition of PVHCRH neurons delays recovery of righting reflex and decreases sevoflurane-induced stress (an apparent decrease in the arousal effect). The authors conclude that PVHCRH neurons "integrate" sevoflurane-induced anesthesia and stress. The authors also claim that their findings show that sevoflurane itself produces a post-anesthesia stress response that is independent of any surgical trauma, such as an incision. In its revised form, the article does not achieve its intended goal and will not have impact on the clinical practice of anesthesiology nor on anesthesiology research.

Strengths:

The manuscript uses targeted manipulation of the PVHCRH neurons with state-of-the-art methods and is technically sound. Also, the number of experiments is substantial.

Weaknesses:

The most significant weaknesses remain: (a) overinterpretation of the significance of their findings (b) the failure to use another anesthetic as a control, (c) a failure to compellingly link their post-sevoflurane measures in mice to anything measured in humans, and (d) limitations in the novelty of the findings. These weaknesses are related to the primary concerns described below:

Concerns about the primary conclusion that PVHCRH neurons integrate the anesthetic effects and post-anesthesia stress response of sevoflurane GA:

It is important to compare the effects of sevoflurane with at least one other inhaled ether anesthetic as one step towards elevating the impact of this paper to the level required for a journal such as eLife. Isoflurane, desflurane, and enflurane are ether anesthetics that are very similar to each other, as well as being similar to sevoflurane. For example, one study cited by the authors (Marana et al. 2013) concludes that there is weak evidence for differences in stress-related hormones between sevoflurane and desflurane, with lower levels of cortisol and ACTH observed during the desflurane intraoperative period. It is important to determine whether desflurane activates PVHCRH neurons in the post-anesthesia period, and whether this is accompanied by excess grooming in the mice because this will distinguish whether the effects of sevoflurane generalize to other inhaled anesthestics, or, alternatively, relate to unique idiosyncratic properties of this gas that may not be a part of its anesthetic properties.

Concerns about the clinical relevance of the experiments:

In anesthesiology practice, perioperative stress observed in patients is more commonly related to the trauma of the surgical intervention, with inadequate levels of antinociception or unconsciousness intraoperatively and/or poor post-operative pain control. The authors seem to be suggesting that the sevoflurane itself is causing stress because their mice receive sevoflurane but no invasive procedures, but there is no evidence of sevoflurane inducing stress in human patients. It is important to know whether sevoflurane effectively produces behavioral stress in the recovery room in patients that could be related to the putative stress response (excess grooming) observed in mice. For example, in surgeries or procedures which required only a brief period of unconsciousness that could be achieved by administering sevoflurane alone (comparable to the 30 min administered to the mice), is there clinical evidence of post-operative stress? It is also important to describe a rationale for using a 30 min sevoflurane exposure. What proportion of human surgeries using sevoflurane use exposure times that are comparable to this?

It is the experience of one of the reviewers that human patients who receive sevoflurane as the primary anesthetic do not wake up more stressed than if they had had one of the other GABAergic anesthetics. If there were signs of stress upon emergence (increased heart rate, blood pressure, thrashing movements) from general anesthesia, this would be treated immediately. The most likely cause of post-operative stress behaviors in humans is probably inadequate anti-nociception during the procedure, which translates into inadequate post-op analgesia and likely delirium. It is the case that children receiving sevoflurane do have a higher likelihood of post-operative delirium. Perhaps the authors' studies address a mechanism for delirium associated with sevoflurane, but this is barely mentioned. Delirium seems likely to be the closest clinical phenomenon to what was studied. As noted by the Besnier et al (2017) article cited by the authors, surgery can elevate postoperative glucocorticoid stress hormones, but it generally correlates with the intensity of the surgical procedure. Besnier et al also note the elevation of glucocorticoids is generally considered to be adaptive. Thus, reducing glucocorticoids during surgery with sevoflurane may hamper recovery, especially as it relates to tissue damage, which was not measured or considered here. This paper only considers glucocorticoid release as a negative factor, which causes "immunosuppression", "proteolysis", and "delays postoperative recovery and...leads to increased morbidity".

It is also the case that there are explicit published findings showing that mild and moderate surgical procedures in children receiving sevoflurane (which might be the closest human proxy to the brief 30 minute sevoflurane exposure used here) do not have elevated cortisol (Taylor et al, J Clin Endocrinol Metab, 2013). This again raises the question of whether the enhanced grooming or elevated corticosterone observed in the mice here has any relevance to humans.

Concerns about the novelty of the findings:

The key finding here is that CRH neurons mediate measures of arousal, and arousal modulates sevoflurane anesthesia induction and recovery. However, CRH is associated with arousal in numerous studies. In fact, the authors' own work, published in eLife in 2021, showed that stimulating the hypothalamic CRH cells lead to arousal and their inhibition promoted hypersomnia. In both papers the authors use fos expression in CRH cells during a specific event to implicate the cells, then manipulate them and measure EEG responses. In the previous work, the cells were active during wakefulness; here- they were active in the awake state the follows anesthesia (Figure 1). Thus, the findings in the current work are incremental and not particularly impactful. Claims like "Here, a core hypothalamic ensemble, corticotropin-releasing hormone neurons in the paraventricular nucleus of the hypothalamus, is discovered" are overstated. PVHCRH cell populations were discovered in the 1980s. Suggesting that it is novel to identify that hypothalamic CRH cells regulate post-anesthesia stress is unfounded as well: this PVH population has been shown over four decades to regulate a plethora of different responses to stress. Anesthesia stress is no different. Their role in arousal is not being discovered in this paper. Even their role in grooming is not discovered in this paper.

The activation of CRH cells in PVH has already been shown to result in grooming by Jaideep Bains (a paper cited by the authors). Thus, the involvement of these cells in this behavior is not surprising. The authors perform elaborate manipulations of CRH cells and numerous analyses of grooming and related behaviors. For example, they compare grooming and paw licking after anesthesia with those after other stressors such as forced swim, spraying mice with water, physical attack and restraint. The authors have identified a behavioral phenomenon in a rodent model that does not have a clear correlation with a behavior state observed in humans during the use of sevoflurane as part of an anesthetic regimen. The grooming behaviors are not a model of the emergence delirium or the cognitive dysfunction observed commonly in patients receiving sevoflurane for general anesthesia. Emergence delirium is commonly seen in children after sevoflurane is used as part of general anesthesia and cognitive dysfunction is commonly observed in adults-particularly the elderly -- following general anesthesia.

---

## [Author Response]

The following is the authors’ response to the previous reviews.

**Public reviews**
This study describes a group of CRH-releasing neurons, located in the paraventricular nucleus of the hypothalamus, which, in mice, affects both the state of sevoflurane anesthesia and a grooming behavior observed after it. PVHCRH neurons showed elevated calcium activity during the post-anesthesia period. Optogenetic activation of these PVHCRH neurons during sevoflurane anesthesia shifts the EEG from burst-suppression to a seemingly activated state (an apparent arousal effect), although without a behavioral correlate. Chemogenetic activation of the PVHCRH neurons delays sevoflurane-induced loss of righting reflex (another apparent arousal effect). On the other hand, chemogenetic inhibition of PVHCRH neurons delays recovery of righting reflex and decreases sevoflurane-induced stress (an apparent decrease in the arousal effect). The authors conclude that PVHCRH neurons "integrate" sevoflurane-induced anesthesia and stress. The authors also claim that their findings show that sevoflurane itself produces a post-anesthesia stress response that is independent of any surgical trauma, such as an incision. In its revised form, the article does not achieve its intended goal and will not have impact on the clinical practice of anesthesiology nor on anesthesiology research.

Thanks for the reviews. Please see our responses to the following comments.

Weaknesses:The most significant weaknesses remain:a) overinterpretation of the significance of their findingsb) the failure to use another anesthetic as a control,c) a failure to compellingly link their post-sevoflurane measures in mice to anything measured in humans, andd) limitations in the novelty of the findings. These weaknesses are related to the primary concerns described below:Concerns about the primary conclusion that PVHCRH neurons integrate the anesthetic effects and post-anesthesia stress response of sevoflurane GA(1) After revision, their remain multiple places where it is claimed that PVHCRH neurons mediate the anesthetic effects of sevoflurane (impact statement: we explain "how sevoflurane-induced general anesthesia works..."; introduction: "the neuronal mechanisms that mediate the anesthetic effects...of sevoflurane GA remain poorly understood" and "PVHCRH neurons may act as a crucial node integrating the anesthetic effect and stress response of sevoflurane").The manuscript simply does not support these statements. The authors show that a short duration exposure to sevoflurane inhibits PVHCRH neurons, but this is followed by hyperexcitability of these neurons for a short period after anesthesia is terminated. They show that the induction and recovery from sevoflurane anesthesia can be modulated by PVHCRH neuronal activity, most likely through changes in brain state (measured by EEG). They also show that PVHCRH neuronal activity modulates corticosterone levels and grooming behavior observed post-anesthesia (which the authors argue are two stress responses).These two things (effects during anesthesia and effects post-anesthesia)may be mechanistically unrelated to each other. None of these observations relate to the primary mechanism of action for sevoflurane. All claims relating to "anesthetic effects" should be removed. Even the term "integration" seems wrong-it implies the PVH is combining information about the anesthetic effect and post-anesthesia stress responses.

As requested, we have removed all claims related to ‘anesthetic effects’ or ‘integration’. Please see the revised manuscript.

(2) lt is important to compare the effects of sevoflurane with at least one other inhaled ether anesthetic as one step towards elevating the impact of this paper to the level required for a journal such as eLife. Isoflurane, desflurane, and enflurane are ether anesthetics that are very similar to each other, as well as being similar to sevoflurane. For example, one study cited by the authors (Marana et al.2013) concludes that there is weak evidence for differences in stress-related hormones between sevoflurane and desflurane, with lower levels of cortisol and ACTH observed during the desflurane intraoperative period. It is important to determine whether desflurane activates PVHCRH neurons in the post-anesthesia period, and whether this is accompanied by excess grooming in the mice, because this will distinguish whether the effects of sevoflurane generalize to other inhaled anesthestics, or, alternatively, relate to unique idiosyncratic properties of this gas that may not be a part of its anesthetic properties.

Thanks for your insightful comments and suggestions. Regarding your request for additional experiments, we acknowledge the value they could add to our study. However, investigating whether the effects of sevoflurane generalize to other inhaled anesthetics is beyond the scope of our current study. There is evidence indicating the prevalence of anesthetic stress caused by inhaled ether anesthetics1,2. The post-anesthesia stress-related behaviors caused by sevoflurane administration are reminiscent of delirium observed clinically. Notably, studies have shown that the use of desflurane for maintenance of anesthesia did not significantly affect the incidence or duration of delirium compared to sevoflurane administration3. This suggests that our observations likely represent a generalized response to inhaled ether anesthetic rather than being specific to sevoflurane.

Concerns about the clinical relevance of the experimentsIn anesthesiology practice, perioperative stress observed in patients is more commonly related to the trauma of the surgical intervention, with inadequate levels of antinociception or unconsciousness intraoperatively and/or poor post-operative pain control. The authors seem to be suggesting that the sevoflurane itself is causing stress because their mice receive sevoflurane but no invasive procedures, but there is no evidence of sevoflurane inducing stress in human patients. It is important to know whether sevoflurane effectively produces behavioral stress in the recovery room in patients that could be related to the putative stress response (excess grooming) observed in mice. For example, in surgeries or procedures which required only a brief period of unconsciousness that could be achieved by administering sevoflurane alone (comparable to the 30 min administered to the mice), is there clinical evidence of post-operative stress? It is also important to describe a rationale for using a 30 min sevoflurane exposure. What proportion of human surgeries using sevoflurane use exposure times that are comparable to this?It is also the case that there are explicit published findings showing that mild and moderate surgical procedures in children receiving sevoflurane (which might be the closest human proxy to the brief 30 minutes sevoflurane exposure used here) do not have elevated cortisol (Taylor et al, J Clin Endocrinol Metab, 2013). This again raises the question of whether the enhanced grooming or elevated corticosterone observed in the mice here has any relevance to humans.

Thanks for the comments. Most ear, nose, and throat surgeries in children involve a short period of anesthesia with sevoflurane alone4-6, which is similar to the 30-minute exposure in our mouse study. In clinical settings, emergence delirium and agitation are common in young children undergoing sevoflurane anesthesia7, often accompanied by troublesome excitation phenomena during induction and awakening8. These clinical observations align with the post-operative stress response (e.g., excessive grooming) we identified in our study.

It is the experience of one of the reviewers that human patients who receive sevoflurane as the primary anesthetic do not wake up more stressed than if they had had one of the other GABAergic anesthetics. If there were signs of stress upon emergence (increased heart rate, blood pressure, thrashing movements) from general anesthesia, this would be treated immediately. The most likely cause of post-operative stress behaviors in humans is probably inadequate anti-nociception during the procedure, which translates into inadequate post-op analgesia and likely delirium. It is the case that children receiving sevoflurane do have a higher likelihood of post-operative delirium. Perhaps the authors' studies address a mechanism for delirium associated with sevoflurane, but this is barely mentioned. Delirium seems likely to be the closest clinical phenomenon to what was studied. As noted by the Besnier et al (2017) article cited by the authors, surgery can elevate postoperative glucocorticoidstress hormones, but it generally correlates with the intensity of the surgical procedure. Besnier et al also note the elevation of glucocorticoids is generally considered to be adaptive. Thus, reducing glucocorticoids during surgery with sevoflurane may hamper recovery, especially as it relates to tissue damage, which was not measured or considered here. This paper only considers glucocorticoid release as a negative factor, which causes "immunosuppression", "proteolysis", and "delays postoperative recovery and leads to increased morbidity".

Thanks for the comments. We agree that the post-anesthetic stress behaviors mentioned in our manuscript are similar to the clinical phenomenon of delirium, which were defined in Cheng Li’s study as ‘sevoflurane-induced post-operative delirium’9. Therefore, we conducted additional behavioral tests for cognitive function, including the Y-maze and novel object recognition test, in mice administrated 30-minute sevoflurane anesthesia. The results demonstrate that chemogenetic inhibition of PVHCRH neurons ameliorated the short-term memory impairment in mice exposed to 30-minute sevoflurane GA (Figure 7-figure supplement 9), suggesting PVHCRH neurons may involve in modulating sevoflurane-induced postoperative delirium.

Concerns about the novelty of the findings:The key finding here is that CRH neurons mediate measures of arousal, and arousal modulates sevoflurane anesthesia induction and recovery. However, CRH is associated with arousal in numerous studies. In fact, the authors' own work, published in eLife in 2021, showed that stimulating the hypothalamic CRH cells lead to arousal and their inhibition promoted hypersomnia. In both papers the authors use fos expression in CRH cells during a specific event to implicate the cells, then manipulate them and measure EEG responses. In the previous work, the cells were active during wakefulness; here- they were active in the awake state the follows anesthesia (Figure1). Thus, the findings in the current work are incremental and not particularly impactful. Claims like "Here, a core hypothalamic ensemble, corticotropin-releasing hormone neurons in the paraventricular nucleus of the hypothalamus, is discovered" are overstated. PVHCRH cell populations were discovered in the 1980s. Suggesting that it is novel to identify that hypothalamic CRH cells regulate post-anesthesia stress is unfounded as well: this PVH population has been shown over four decades to regulate a plethora of different responses to stress. Anesthesia stress is no different. Their role in arousal is not being discovered in this paper. Even their role in grooming is not discovered in this paper.

Thanks for the comments. As requested, we have revised our manuscript by removing overstated sentences. Please see the revised manuscript. In terms of novelty, our study reveals that PVHCRH neurons are implicated not only in the induction and emergence of sevoflurane general anesthesia but also in sevoflurane-induced post-operative delirium. This finding represents a novel contribution to the field, as it has not been previously reported by other studies.

The activation of CRH cells in PVH has already been shown to result in grooming by Jaideep Bains (a paper cited by the authors). Thus, the involvement of these cells in this behavior is not surprising. The authors perform elaborate manipulations of CRH cells and numerous analyses of grooming and related behaviors. For example, they compare grooming and paw licking after anesthesia with those after other stressors such as forced swim, spraying mice with water, physical attack and restraint. The authors have identified a behavioral phenomenon in a rodent model that does not have a clear correlation with a behavior state observed in humans during the use of sevoflurane as part of an anesthetic regimen. The grooming behaviors are not a model of the emergence delirium or the cognitive dysfunction observed commonly in patients receiving sevoflurane for general anesthesia. Emergence delirium is commonly seen in children after sevoflurane is used as part of general anesthesia and cognitive dysfunction is commonly observed in adults-particularly the elderly-- following general anesthesia. No features of delirium or cognitive dysfunction are measured here.

As requested, behavioral tests for cognitive function have been conducted and displayed in Figure 7-figure supplement 9.

Other concerns:In Figure 2, cFos was measured in the PVH at different points before, during and after sevoflurane. The greatest cFos expression was seen in Post 2, the latest time point after anesthesia. However, this may simply reflect the fact that there is a delay between activity levels and expression of cFos (as noted by the authors, 2-3 hours). Thus, sacrificing mice 30 minutes after the onset of sevoflurane application would be expected to drive minimal cFos expression, and the cFos observed at 30 minutes would not accurately reflect the activity levels during the sevoflurane. Also, the authors state that the hyperactivity, as measured by cFos, lasted "approximately 1 hours before returning to baseline", but there is no data to support this return to baseline.

Thanks for the comments. We apologize that the protocol we used for c-fos staining may not accurately reflect the activity levels, so we have removed Figure 2F. The sentence ‘lasted approximately 1 hours before returning to baseline’ refers to the calcium signal but not c-fos level.

In Figure 7, the number of animals appears to change from panel to panel even though they are supposed to show animals from the same groups. For example, cort was measured in only 3 saline-treated O2 animals (Fig 7E), but cFos and CRH were assessed in 4 (Fig C,D). Similarly, grooming time and time spent in open arms was measured in 6 saline-treated O2 controls (Fig 7F, H) but central distance was measured in 8(Fig 7G). There are other group number discrepancies in this figure--the number of data points in the plots do not match what is reported in the legend for numerous groups. Similarly, Figure 4 has a mismatch between the Ns reported in the legend and the number of points plotted per bar. For example, there were 10 animals in the hM3Di group; all are shown for the LORR and time to emergence plots, but only8 were used for time to induction. The legends reported N=7 for the mCherry group, yet 9 are shown for the time to emergence panel. No reason for exclusions is cited. These figures (and their statistics) should be corrected.

Thanks for the comments. We have rechecked and corrected our figures and illustrations in the revised manuscript.

**Recommendations for the authors:**
In Figure 6, the BSR pre-stim data points for panels F and H look exactly identical, even though these data are from two different sets of mice. It seems likely that one of these panels is not depicting the correct pre-stim data points. Please check this.

Thanks for the comments. We have corrected this mistake.

General anesthesia is a combination of behavioral and physiological states induced and maintained primarily by pharmacologic agents. The authors do not provide a definition of general anesthesia.

Thanks for the advice. We have added the definition of general anesthesia in the introduction part.

The first sentence of the abstract closely resembles the first sentence of the abstract of Brown，Purdon and Van Dort,Annu. Rev. Neurosci. 2011,34:601-28 yet, there is no citation.

Thanks for the comments. We have revised the first sentence.

ln the Discussion, the authors cite the research on circuitry that is relevant for emergence from general anesthesia. Conspicuously missing from this section of the paper is the large body of work by Solt and colleagues which has demonstrated that dopamine agonists (such as methylphenidate), electrical stimulation of the ventral tegmental area and optogenetic stimulation of the D1 neurons in the ventral tegmental area can hasten emergence from general anesthesia. Also omitted is the work of Kelzand colleagues and a discussion of neural inertia.

Thanks for the suggestions. We have added these citations as requested.

As regards the weaknesses of p-values for reporting the results of scientific studies, l offer the following reference to the authors. Ronald L. Wasserstein & Nicole A.Lazar (2016)The ASA Statement on p-Values: Context, Process, and Purpose, The American Statistician,70:2,129- 133, DOl:10.1080/00031305.2016.1154108

Thanks for the suggestions. We have revised the manuscript as requested.

The methods for the CRF antibody are unclear. It was previously suggested that the antibody be validated (for example, show an absence of immunostaining with CRF knockdown) because the concentration of antiserum (1:800) is quite high, suggesting either the antibody is not potent or (more concerning) not specific. The methods also indicated that colchicine was infused ICV prior to perfusion for staining of cFos and CRF, but no surgical methods are described that would enable ICV infusion, and it is not clear why colchicine was used. Please clarify.

The anti-CRF antibody is validated by other studies11,12. F For CRF immunostaining, animals' brains were pre-treated with intraventricular injections of colchicine (20 μg in 500 nL saline) 24 hours before perfusion to inhibit fast axonal transport13,14. Additional details regarding these methods have been included in the Method section of the revised manuscript.

**Editor's note:**
Full statistical reporting including exact p-values alongside summary statistics (test statistic and df) and 95% confidence intervals is lacking.

Thanks for the suggestions. We have added full statistical reporting in the revised manuscript as requested.

Reference

(1) Marana, E. *et al.* Desflurane versus sevoflurane: a comparison on stress response. *Minerva Anestesiol*
**79**, 7-14 (2013).

(2) Yang, L., Chen, Z. & Xiang, D. Effects of intravenous anesthesia with sevoflurane combined with propofol on intraoperative hemodynamics, postoperative stress disorder and cognitive function in elderly patients undergoing laparoscopic surgery. *Pak J Med Sci*
**38**, 1938-1944, doi:10.12669/pjms.38.7.5763 (2022).

(3) Driscoll, J. N. *et al.* Comparing incidence of emergence delirium between sevoflurane and desflurane in children following routine otolaryngology procedures. *Minerva Anestesiol*
**83**, 383-391, doi:10.23736/s0375-9393.16.11362-8 (2017).

(4) Galinkin, J. L. *et al.* Use of intranasal fentanyl in children undergoing myringotomy and tube placement during halothane and sevoflurane anesthesia. *Anesthesiology*
**93**, 1378-1383, doi:10.1097/00000542-200012000-00006 (2000).

(5) Greenspun, J. C., Hannallah, R. S., Welborn, L. G. & Norden, J. M. Comparison of sevoflurane and halothane anesthesia in children undergoing outpatient ear, nose, and throat surgery. *J Clin Anesth*
**7**, 398-402, doi:10.1016/0952-8180(95)00071-o (1995).

(6) Messieha, Z. Prevention of sevoflurane delirium and agitation with propofol. *Anesth Prog*
**60**, 67-71, doi:10.2344/0003-3006-60.3.67 (2013).

(7) Shi, M. *et al.* Dexmedetomidine for the prevention of emergence delirium and postoperative behavioral changes in pediatric patients with sevoflurane anesthesia: a double-blind, randomized trial. *Drug Des Devel Ther*
**13**, 897-905, doi:10.2147/dddt.S196075 (2019).

(8) Veyckemans, F. Excitation and delirium during sevoflurane anesthesia in pediatric patients. *Minerva Anestesiol*
**68**, 402-405 (2002).

(9) Xu, Y., Gao, G., Sun, X., Liu, Q. & Li, C. ATPase Inhibitory Factor 1 Is Critical for Regulating Sevoflurane-Induced Microglial Inflammatory Responses and Caspase-3 Activation. *Front Cell Neurosci*
**15**, 770666, doi:10.3389/fncel.2021.770666 (2021).

(10) Friedman, E. B. *et al.* A conserved behavioral state barrier impedes transitions between anesthetic-induced unconsciousness and wakefulness: evidence for neural inertia. *PLoS One*
**5**, e11903, doi:10.1371/journal.pone.0011903 (2010).

(11) Giardino, W. J. *et al.* Parallel circuits from the bed nuclei of stria terminalis to the lateral hypothalamus drive opposing emotional states. *Nat Neurosci*
**21**, 1084-1095, doi:10.1038/s41593-018-0198-x (2018).

(12) Yeo, S. H., Kyle, V., Blouet, C., Jones, S. & Colledge, W. H. Mapping neuronal inputs to Kiss1 neurons in the arcuate nucleus of the mouse. *PLoS One*
**14**, e0213927, doi:10.1371/journal.pone.0213927 (2019).

(13) de Goeij, D. C. *et al.* Repeated stress-induced activation of corticotropin-releasing factor neurons enhances vasopressin stores and colocalization with corticotropin-releasing factor in the median eminence of rats. *Neuroendocrinology*
**53**, 150-159, doi:10.1159/000125712 (1991).

(14) Yuan, Y. *et al.* Reward Inhibits Paraventricular CRH Neurons to Relieve Stress. *Curr Biol*
**29**, 1243-1251.e1244, doi:10.1016/j.cub.2019.02.048 (2019).